# Assimilating realistically simulated wide-swath altimeter observations in a high-resolution shelf-seas forecasting system

Robert R. King[1] and Matthew J. Martin[1]

[1]Met Office, Exeter, UK

**Correspondence:** Robert R. King (robert.r.king@metoffice.gov.uk)

**Abstract.**

The impact of assimilating simulated wide-swath altimetry observations from the upcoming SWOT mission is assessed using Observing System Simulation Experiments (OSSEs). These experiments use the Met Office 1.5 km resolution North-West European Shelf analysis and forecasting system. In an effort to understand the importance of future work to account for correlated errors in the data assimilation scheme, we simulate SWOT observations with and without realistic correlated errors. These are assimilated in OSSEs along with simulated observations of the standard observing network, also with realistic errors added.

It was found that while the assimilation of SWOT observations without correlated errors reduced the RMSE (Root Mean Squared Error) in sea surface height (SSH) and surface current speeds by up to 20%, the inclusion of correlated errors in the observations degraded both the SSH and surface currents, introduced an erroneous increase in the mean surface currents, and degraded the sub-surface temperature and salinity. While restricting the SWOT data to the inner half of the swath and applying observation averaging with a 5 km radius negated most of the negative impacts, it also severely limited the positive impacts. To realise the full benefits in the prediction of the ocean mesoscale offered by wide-swath altimetry missions it is crucial that methods to ameliorate the effects of correlated errors in the processing of the SWOT observations and to account for the correlated errors in the assimilation are implemented.

## 1 Introduction

Satellite altimeter observations of the sea surface height (SSH) have been available for over 25 years and are routinely assimilated in operational global ocean analysis and forecasting systems (e.g. Waters et al., 2015; Oke et al., 2015; Lellouche et al., 2018). The global coverage of a constellation of altimeters provides constraints on the modelled SSH and geostrophic velocity

field, including the position and strength of mesoscale eddies. The SSH data can also be used to adjust the sub-surface density
structure (e.g. Weaver et al., 2005). The complementarity of satellite altimetry and in situ measurements of the temperature
and salinity was demonstrated by Lea et al. (2014) where it was shown that assimilating SSH observations improves the model
forecast of other variables even where more direct measurements are available.

Due to the large gaps between tracks and long repeat cycles (10–35 days), existing along-track altimeter observations pro-
vide limited spatial and temporal resolution compared with the scales of the ocean mesoscale and sub-mesoscale dynamics.
Various studies have attempted to quantify the impact of different numbers of standard along-track (nadir) satellite altimeters
on ocean analysis and forecasting systems. For instance, Jacobs et al. (2014) showed that even with 4 nadir altimeters, while
the SSH itself can be reasonably well constrained, other parameters related to the prediction of frontal positions were not well
constrained. Although along-track observations can have a sampling frequency of ∼7 km, various sources of noise limit the
feature resolution to ∼50 km (Dufau et al., 2016). This is already sub-optimal for the initialisation of mesoscale features in
the current generation of operational ocean prediction systems. With the move to higher-resolutions, this disparity between the
observed and modelled scales will increase.

The operational shelf-seas short-range prediction system at the Met Office uses an eddy-resolving 1.5 km model in the
North-West European Shelf (NWS; Tonani et al., 2019) and delivers products to various users, including the Copernicus
Marine Service (https://marine.copernicus.eu). The current operational system assimilates altimeter observations only in the
deep water (>700 m depth) regions of the domain (King et al., 2018). Satellite and in situ sea surface temperature (SST) data,
and in situ profiles of temperature and salinity, are assimilated throughout the domain. Given the sparseness of the in situ
observations over this region (De Mey-Frémaux et al., 2019), satellite altimeters could provide potentially useful information
on the vertically-integrated water column across the domain. However, if there are insufficient in situ observations to constrain
the sub-surface density structure, the assimilation of SSH observations can lead to unrealistic changes to the sub-surface density
(Lea et al., 2014) and an overall degradation in forecast quality. As well as inferring corrections to sub-surface properties, SSH
data from altimeters could help to constrain processes such as tides and storm surges which are represented in the model and
are observed by altimeters. However, current operational systems are not optimised to make full use of altimeter observations
in constraining these dynamical processes. Furthermore, the sampling of these processes by existing nadir altimeters is not
currently sufficient to adequately constrain them in the NWS region.
Upcoming wide-swath altimetry missions, such as SWOT (Durand et al., 2010) and COMPIRA (Uematsu et al., 2013),
will provide a step-change in the ability to observe ocean mesoscale features by measuring the SSH along a wide swath and
providing 2D measurements of the ocean surface topography. The SWOT mission will have a 120 km wide swath with a 20 km
central gap combined with a nadir altimeter and aims to provide an effective resolution of ∼15 km (Morrow et al., 2018), but
with sampling of ∼2 km within the swath. As with the current nadir altimeters, the long repeat cycle of SWOT (21 days, 11 day
average repeat time) means that although high-resolution 2D maps of the SSH will be available, there will be long periods
where each location is unobserved.

Wide-swath altimetry observations will be subject to instrument noise and large correlated geophysical and instrumental
errors (Gaultier et al., 2016). Although strategies to reduce the magnitude of some of these errors may be developed before

routine dissemination of the observations, this presents a challenge for data assimilation schemes. Most operational data assimilation schemes assume observation errors to be uncorrelated. Common methods to ameliorate the effects of correlated observation errors are to thin or average nearby observations, or to increase the observation error variances to avoid over-fitting the data (e.g., Fowler et al., 2018). All these common methods suffer from the fact that they do not allow the data assimilation to make best use of the small-scale information contained in the observations. There has been work to extend data assimilation schemes to explicitly account for correlated observation errors. For example, Bédard and Buehner (2020) used a 1D model to demonstrate the combination of direct and spatial difference observations to account for observation error correlations, while Ruggiero et al. (2016) and Guillet et al. (2019) parameterised the observation error covariances using the diffusion equation to avoid the need to invert a non-diagonal covariance matrix. On the other hand, Metref et al. (2020) used the correlated-error reduction (CER) method to account for correlated errors when assimilating SWOT observations in a one-and-a-half layer quasi-geostrophic model. These methods are fairly immature in terms of implementation in operational systems, but should be a focus for implementation over the coming years.

A number of studies have also examined the potential impact of assimilating SWOT observations in operational-like ocean analysis and forecasting systems. Carrier et al. (2016) demonstrated positive impacts from assimilating simulated SWOT observations with the full correlated errors into a 6 km resolution regional model, where the SWOT observations were thinned to ∼20 km separation between observations. D'Addezio et al. (2019) found that assimilating simulated SWOT observations with only uncorrelated errors in a 1 km model decreased the minimum constrained length-scale, but did not significantly improve area-averaged errors. On the other hand, Bonaduce et al. (2018) and Li et al. (2019) find significant improvements in regional models from assimilating SWOT observations, as do Benkiran et al. (2021) and Tchonang et al. (2021) in a global $1/12°$ model, but in each case the simulated observations did not include correlated errors. The limitations of ignoring the effects of one of the expected sources of correlated errors was explored by Le Henaff et al. (2008) using a barotropic model of the North Sea.

The aim of this work is to assess the benefit of assimilating realistic SWOT-like observations within our current shelf-seas forecasting system and to examine some simple strategies to address the impact of the large correlated errors expected in realistic wide-swath altimeter observations. Our intention is initially not to adapt the data assimilation scheme to account for correlated errors. Rather we aim to quantify the impact that might be expected when mitigating for their effects and to understand the issues that the full set of expected correlated errors bring about when used in the assimilation. Here we have investigated the impact of restricting the SWOT data assimilated to the inner half of the swath (to discard the largest errors) and the impact of applying median-averaging of the observations with different averaging radii.

We use Observing System Simulation Experiments (OSSEs), as described in Section 2, to perform idealised tests of the assimilation of SWOT observations alongside the existing network of satellite and in situ observations in the 1.5 km model of the North-West European Shelf mentioned earlier. This system uses the NEMO model together with the NEMOVAR data assimilation system which are described in Section 3. The details of the simulation of the observations assimilated in the OSSE are presented in Section 4. The results of the experiments are presented in Section 5 and a discussion of the main points to come out of the results is given in Section 6. Finally our conclusions are presented in Section 7.

**Table 1.** Experimental set-up

| Experiment | Standard Observations | Swath width | SWOT errors | Superob radius | Period | | Surface forcing |
|---|---|---|---|---|---|---|---|
| Nature Run | - | - | - | - | 01 Jan 2017 | 31 Dec 2018 | MetUM |
| Free Run | - | - | - | - | 01 Jul 2017 | 31 Dec 2018 | ECMWF IFS |
| Control | Y | - | - | - | 01 Jul 2017 | 30 Jun 2018 | ECMWF IFS |
| HalfSWOT | Y | Half width | All | - | 01 Jul 2017 | 30 Jun 2018 | ECMWF IFS |
| SWOT | Y | Full width | All | - | 01 Jul 2017 | 30 Jun 2018 | ECMWF IFS |
| LowErrSWOT | Y | Full width | Uncorrelated | - | 01 Jul 2017 | 30 Jun 2018 | ECMWF IFS |
| HalfSWOT_5km | Y | Half width | All | 5 km | 01 Jul 2017 | 30 Jun 2018 | ECMWF IFS |
| HalfSWOT_20km | Y | Half width | All | 20 km | 01 Jul 2017 | 30 Jun 2018 | ECMWF IFS |
| SWOT_5km | Y | Full width | All | 5 km | 01 Jul 2017 | 30 Jun 2018 | ECMWF IFS |
| SWOT_20km | Y | Full width | All | 20 km | 01 Jul 2017 | 30 Jun 2018 | ECMWF IFS |

## 2 Experiment Design

An OSSE is used to assess the impact on ocean analyses and forecasts of potential changes to the current observing system and consists of several components: a Nature Run, observations simulated from the Nature Run, and additional OSSE runs using a different model set-up into which the simulated observations are assimilated. Examples of best practices for running OSSEs include Hoffman and Atlas (2016) and Halliwell et al. (2017).

The Nature Run provides the truth against which other experiments are assessed and from which we sample observations. To draw useful conclusions about changes to the observing system, the Nature Run needs to be a realistic representation of the real ocean. This is generally accomplished by using the highest resolution model available. To emulate an operational ocean analysis system, realistic errors must be added to the simulated observations sampled from the Nature Run. In Section 4, we detail how we have simulated the observations for each type assimilated in these experiments.

For the OSSE runs, the ocean model must differ in enough respects that there is sufficient error growth between the Nature Run and OSSEs to emulate the differences between an operational system and the real ocean. Ideally, an OSSE would use a separate lower resolution ocean model with different parameterisations, surface and lateral boundaries, and be initialised from a different (though realistic) state (Halliwell et al., 2014). A Control Run, in which some standard network of observations is assimilated, then provides a baseline against which to determine the additional benefit (or detriment) of introducing assimilation of additional observations. Further runs of this model assimilating extra observations in addition to the standard set are used to assess the impact of the additional observations.

For this work, we have employed the same high resolution shelf-seas model with the same lateral boundary conditions in all experiments: a lower resolution model would be unable to resolve the scales which are resolved in the higher resolution model and are expected to be observed with wide-swath altimeter missions. Also, we do not have access to a long run of a

realistic higher resolution model over the domain of interest. To introduce realistic errors, we have forced the Nature Run and the OSSEs with different atmospheric fields (detailed in Section 3.1) and have selected initial conditions valid at the correct time of year, but taken from different existing runs. In Section 3.1.1, we will compare the differences in free-running versions of the Nature Run and OSSE models. A Free Run of the OSSE model, although not strictly necessary, allows an assessment of the errors introduced by the differences between the two models before introducing assimilation. This gives a baseline for the error reduction that assimilation can provide.

Here we have performed a Nature Run, a Free Run, a Control Run, and 7 further experiments (as summarised in Table 1). The Nature Run covered the period 01 Jan 2017 – 31 Dec 2018, while the Free Run started from 01 July 2017 and ran until 31 Dec 2018. The additional 6 months at the start of the Nature Run was to allow the effects of the differing surface forcing to spin-up as all experiments were initialised from runs which had been forced by ECMWF (European Centre for Medium Range Weather Forecasts) fluxes (as used in the OSSE runs). The 8 OSSE runs all cover the period 01 July 2017 – 30 June 2018. The Nature and Free runs do not assimilate any observations, but the Control Run assimilates simulated Sea Surface Temperature (SST) observations, in situ temperature and salinity (T/S) profiles, and nadir Sea Level Anomaly (SLA) observations from 4 satellite altimeters representative of the current altimeter constellation (Jason-3, AltiKa, Sentinal-3A, and Sentinal-3B). The SWOT satellite, which will have a 120 km wide swath with a central 20 km gap, will also carry a nadir altimeter which will observe along the track in the centre of this swath. All of the OSSEs which include SWOT observations here also include the SWOT nadir altimeter.

In addition to the Control experiment, seven further OSSEs (detailed in Table 1) were performed assimilating the standard set of observations assimilated in the Control run plus additional simulated SWOT observations. In the SWOT experiment, we assimilated the full SWOT data with the full correlated errors, while in the HalfSWOT experiment we assimilated only the inner half of the SWOT swath (i.e., 10–35 km from the nadir point on both sides of the swath). In the LowErrSWOT experiment we again assimilated the full swath, but in this case applied only the uncorrelated KaRIn (Ka-band Radar Interferometer) errors to the simulated observations, not the full correlated errors included in the SWOT and HalfSWOT experiments. The various components of the errors in SWOT observations are detailed further in Section 4.2. Finally, we repeated the SWOT and HalfSWOT experiments with median-averaging (super-obbing) of the SWOT observations using a 5 km and a 20 km averaging radius.

# 3   Model and Assimilation Scheme

## 3.1   Ocean Models

The operational Met Office North-West European Shelf ocean forecasting system (NWS, Tonani et al., 2019) which delivers forecast products to the Copernicus Marine Environment Monitoring Service (CMEMS) is used as the basis for all experiments in this work. This system couples the NEMO ocean model with the WAVEWATCH III spectral wave model (Saulter et al., 2017; Lewis et al., 2019). The domain (shown in Figure 1) extends from approximately 45°N, 20°W to 63°N,11°E on a rotated pole grid, covering the North-West European Shelf and part of the North-East Atlantic ocean with a horizontal resolution of

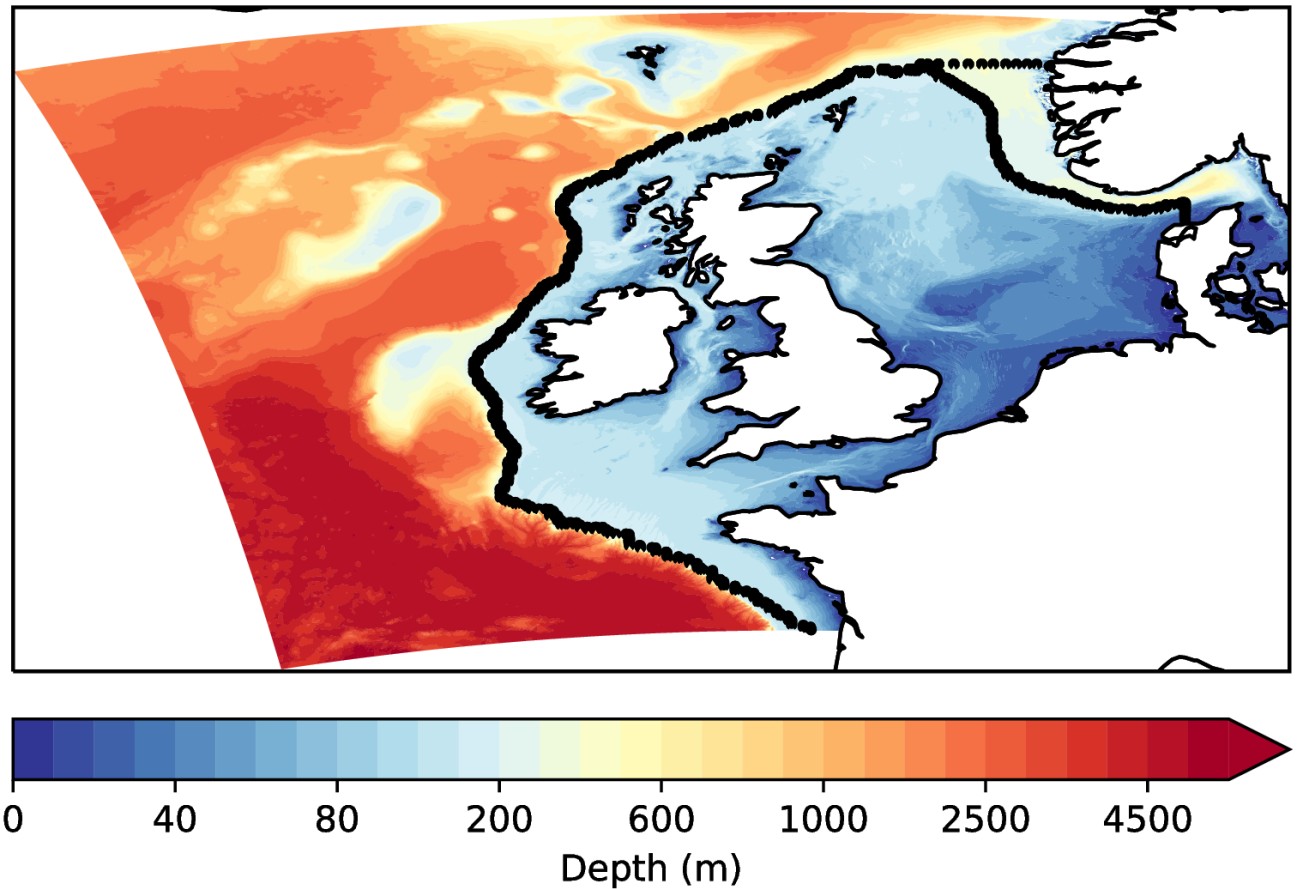

**Figure 1.** The bathymetry of the 1.5 km resolution North-West European shelf system. The boundaries between the off-shelf, on-shelf, and Norwegian Trench regions are marked by a black line. Note that a non-linear colour-sale has been used to highlight the complex bathymetry.

$\sim$1.5 km. This domain has a complex bathymetry, containing regions of shallow water, a continental shelf, and regions of deep ocean (depth$\gtrsim$5000 m). The resulting dynamics produce mesoscale eddies along with seasonally-stratified and well-mixed regions with marked boundaries. The on-shelf and off-shelf regions used in the analysis here are shown in Figure 1. Broadly,
the on-shelf region is defined by the seas around the UK, Ireland and continental Europe where the depth is less than 700 m and excluding the Norwegian Trench and Baltic boundary region.

The ocean model component of the NWS system uses the eddy-resolving 1.5 km Atlantic Margin Model configuration (AMM15, Graham et al., 2018) of NEMO version 3.6 (Madec, 2008). The model employs a non-linear free surface, 51 vertical terrain-following levels and includes 11 tidal constituents, specified on the open boundary via the Flather radiation condition
(Flather, 1981). The AMM15 is nested within the Met Office 1/12° North Atlantic model (Storkey et al., 2010; Blockley et al., 2014) which provides the lateral boundary conditions, except for the Baltic boundary which is taken from the CMEMS Baltic

Sea model (Berg and Poulsen, 2012). The ocean and wave components are coupled hourly using the OASIS3-MCT coupler (Craig et al., 2017).

The operational system is forced at the surface using 3-hourly wind and atmospheric pressure fields, radiative and E-P (evaporation-minus-precipitation) fluxes from the ECMWF operational Integrated Forecasting System. For the OSSE runs we have used the same surface forcing as used in the operational NWS system, but the Nature Run is forced at the surface by the Met Office Unified Model (MetUM) operational global atmospheric model.

### 3.1.1   Assessment of the Nature Run and Free Run

Graham et al. (2018) shows that the AMM15 configuration on which our experiments are based provides a realistic representation of the mean state and variability of the North-West European Shelf. It is able to reproduce the large-scale circulation as well as dynamical features such as mesoscale eddies, frontal jets and internal tides. Although biases are present in the mean surface temperatures and depth of the mixed layer, the model reproduces the variability in the observed long-term average surface temperatures, and the seasonal stratification across the domain.

As mentioned above, the Free Run uses the same ocean model configuration as the Nature Run, but with different atmospheric forcing and a different initial condition to introduce realistic differences between the Nature Run and OSSE runs. Both forcing sets are high-quality atmospheric forecasts with the differences believed to be a fair reflection on the uncertainty in these estimates of the true atmospheric forcing. Similarly, both initial conditions used came from assimilative runs at the correct time of year and the differences reflect the uncertainty given the relative lack of sub-surface observations. As expected, the different surface forcing affects the ocean mixed layer, with the off-shelf average mixed layer being up to $\sim$100 m deeper in the Free Run in winter, but with little difference on-shelf in any season. The mean and standard deviation of the SSH over the year is very similar between the Free Run and the Nature Run, with the SSH standard deviation being $\sim$3% lower on- and off-shelf in the Free Run (compared to the NR). However, as is shown in Figure 2, the bias in the daily mean SSH highlights errors introduced by the differences between the simulations (and which assimilation aims to reduce). The impact of the different initial conditions and surface forcing on the Free Run 3D temperature and salinity biases is shown in Figure 3 for the full 18-month period of these runs. The deeper mixed layer depth (MLD) in the Free Run results in cooler temperatures within the mixed layer (due to broadly similar irradiative fluxes between the atmospheric forcing datasets), but with little difference in the off-shelf temperature below the mixed layer. The large initial salinity bias, particularly off-shelf, remains until near the end of the 18-month runs. The eventual convergence here may be a result of using the same lateral boundaries and river inputs for both runs.

To better understand if the SLA differences between our Nature Run and OSSE runs are similar to what might be expected in an operational system, we have compared the mean and RMS of the SSH increments from the Control and LowErrSWOT experiments with those from a separate experiment assimilating real observations in the same model and over the same time period. We found the biases were near zero in all cases and the RMS of the SSH increments was 1.21 cm in the Control, 1.19 cm in our experiment assimilating real observations, and 1.63 cm in the LowErrSWOT experiment when simulated SWOT observations were included. We believe this demonstrates that our Control run is applying similar increments to an operational

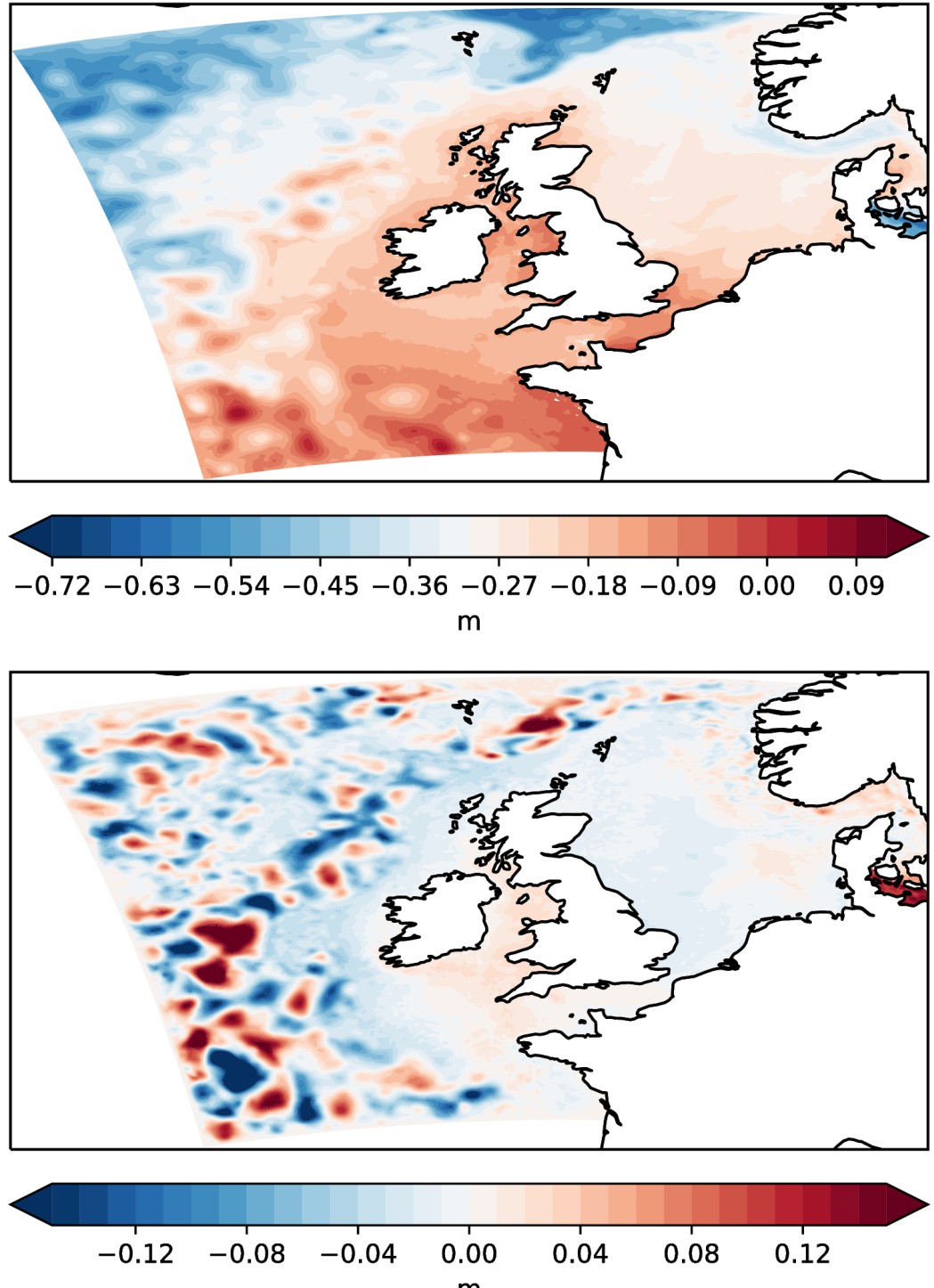

**Figure 2.** Nature Run 25-hour mean SSH for an example day (30 June 2018, top panel) and the difference (Free-Nature bias, bottom panel) in the same field from the Free Run.

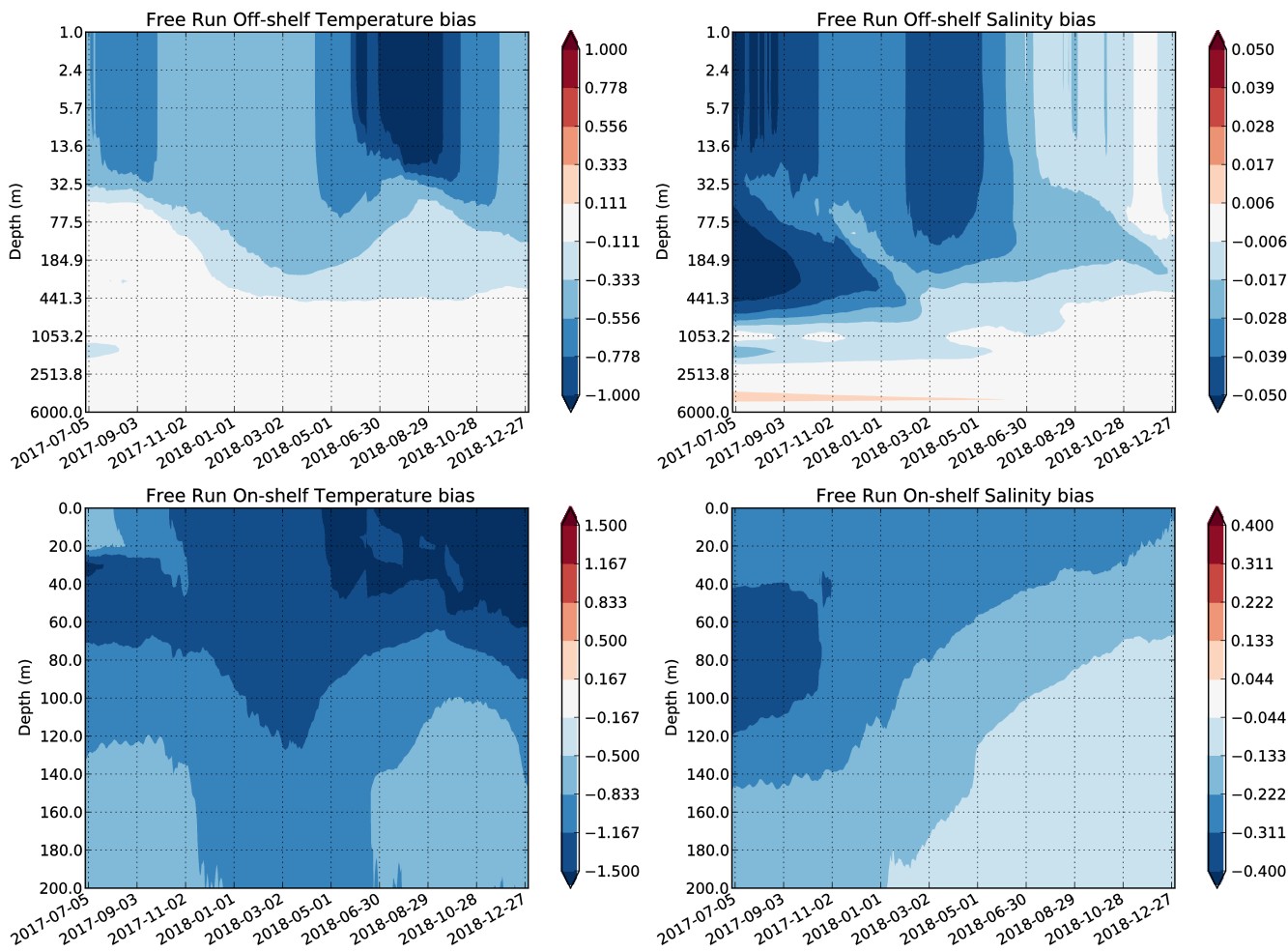

**Figure 3.** Free Run temperature (left) and salinity (right) biases relative to the Nature Run (Free-Nature) for the off-shelf (top) and on-shelf (bottom) regions. Note that the off-shelf plots are shown on a log depth scale.

system assimilating real observations, and the simulated SWOT observations allow more of the SSH variability to be observed and assimilated.

## 3.2 Data Assimilation Scheme

The NEMOVAR data assimilation scheme is used in all configurations of the operational Met Office ocean analysis and forecasting systems. The Met Office NWS ocean forecasting system uses NEMOVAR in its incremental 3D-VAR FGAT (First Guess at Appropriate Time) configuration. This is a multi-variate, multi-lengthscale assimilation scheme developed collaboratively by the Met Office, Centre Européen de Recherche et de Formation Avancée en Calcul Scientifique (CERFACS), ECMWF, and Institut National de Recherche en Informatique et en Automatique (INRIA) (Mogensen et al., 2009, 2012; Waters et al., 2015). The implementation is based on that used in the lower resolution AMM7 system (7 km resolution) as described by King et al. (2018) with modifications as described by Tonani et al. (2019). Two background error correlation length-scales are used in the 1.5 km NWS system: 100 km for the long length-scale and the Rossby radius of deformation for the short length-scale (limited to a minimum of 3 times the grid scale to avoid numerical issues). Each background error scale component has an associated set of spatially and seasonally varying error variances.

An assimilation window of 24 hours is used to assimilate observations including satellite and in situ SST, satellite altimeter observations of the sea level anomaly (SLA), and in situ temperature and salinity profiles from a variety of platforms. The operational system includes a bias correction scheme for SST (While and Martin, 2019) and SLA (Lea et al., 2008) observations, but this is not used here as the observations are simulated directly from the Nature Run. Although we have aimed to add realistic errors to the observations, we have not included any deliberate biases. The assimilation increments are added incrementally over 24 hours through an analysis update cycle (Bloom et al., 1996).

NEMOVAR uses linearised balance relationships to account for correlations between ocean variables as defined by Weaver et al. (2005). The linearised balance relationships between increments to temperature (T), salinity (S), sea surface height ($\eta$), and the u- and v-components of velocity (u,v) can be summarised as follows:

$$\delta T = \delta T,$$
$$\delta S = K_{S,T} \delta T + \delta S_U,$$
$$\delta \eta = K_{\eta,\rho} \delta \rho + \delta \eta_U,$$
$$\delta u = K_{u,p} \delta p + \delta u_U,$$
$$\delta v = K_{v,p} \delta p + \delta v_U. \tag{1}$$

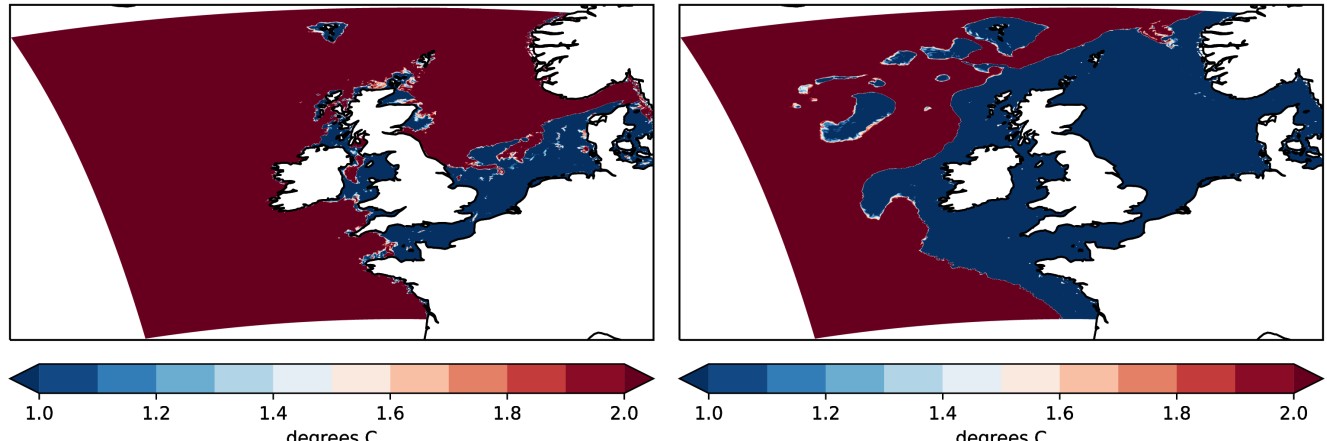

**Figure 4.** The top-to-bottom temperature difference at the end of summer (left, 1 September 2017) and winter (right, 1 March 2018) showing the extent of the domain where no balancing changes are applied to temperature or salinity when assimilating SSH (blue, where $\Delta T < 1^\circ$C) and where the full balance is applied (red, where $\Delta T > 2^\circ$C).

where the subscript $U$ denotes the unbalanced component of the variable. The operators $K_{ij}$ are linearised balance transforms from variable $j$ to $i$. The density ($\rho$) and pressure (p) increments are then defined as:

$$\delta\rho = K_{\rho,T}\delta T + K_{\rho,S}\delta S,$$
$$\delta p = K_{p,\rho}\delta\rho + K_{p,\eta}\delta\eta. \tag{2}$$

The temperature-salinity balance operator ($K_{S,T}$) is a linearised version of the water-mass property conservation scheme of Troccoli and Haines (1999). The SSH balance operator ($K_{\eta,\rho}$) is a linearised version of the Cooper and Haines (1996) scheme in which SSH changes are associated with the lifting and lowering of the entire water column. The velocity balance operators ($K_{uv,p}$) are based on the geostrophic relation, density balance is the linearised equation of state, and the pressure balance is the vertically-integrated hydrostatic equation.

The real SLA observations assimilated in the operational Met Office systems (obtained from CMEMS) are provided with various corrections to remove the effect of tides, atmospheric pressure and winds. These are used in our global model which does not include tides or atmospheric pressure effects. However, our shelf-seas model does include these processes and so the corrections supplied with the SLA observations are not used. However for this work, we have adapted the system to remove tides from the SLA innovations before assimilation which was found in testing to result in slightly reduced errors compared to including the tides in the innovations. This is done by using a 24hour 50minute mean field for the background SSH model field and by using the 25-hour mean fields in the simulation of the SLA observations (as described below in Section 4.2).

We have also removed the previous restrictions on SLA assimilation at high latitudes and in shallow water (depth<700 m) as detailed in King et al. (2018) so that altimeter observations are assimilated throughout the domain. The SSH is adjusted

by applying barotropic (unbalanced) increments to the sea surface height and a baroclinic (balanced) component which acts on the density structure. To avoid making unphysical corrections to the sub-surface temperature and salinity, the full balanced changes to temperature and salinity when assimilating SSH (as detailed in equations 1 and 2) are only applied in regions (shown in Figure 4) where the top-to-bottom temperature difference exceeds 2°C with no balance applied where the difference is less than 1°C and with a linear transition between these thresholds. Broadly, the full balance is applied in the Northern North Sea during June to October, while no balance is applied from December to April in that region. In the tidally-mixed regions such as the Southern North Sea, no balance is applied as the water column is fully mixed throughout the year.

In the experiments reported here, we have used the known observation error variances for the simulated SWOT observations, but have not yet introduced any specific adaptions for the assimilation of wide-swath altimeter observations. In the near future, we will investigate the impact of a number of approaches. Initially, we may test the impact of inflating observation errors for wide-swath altimetry observations to avoid over-fitting the data. However, it will be necessary to investigate methods to account for the correlations directly in the assimilation scheme.

## 4  Simulation of Observations

### 4.1  SST and T/S profiles

To simulate the in situ and level-2 satellite SST observations and in situ T/S profiles assimilated in the operational NWS system, we have used the positions and times of the real observations for the period of the experiments. These include T/S profiles from Argo, XBTs, CTDs, marine mammals, ships and moorings taken from the EN4 database (Good et al., 2013); in situ SST observations from ships, moorings and surface drifters; and satellite SST observations from NOAA-AVHRR, METOP-AVHRR, MSG-SEVIRI, Sentinal3-SLSTR, AMSR-2, and VIIRS. This allowed us to define the position and time of the pseudo-observations to ensure a realistic spatial and temporal coverage. An example of the spatial coverage of daily SST and monthly T/S profile observations is shown in Figure 5. This demonstrates the excellent temporal and spatial sampling of SST observations (with $\sim 100,000$ observations per day) in contrast to the sparse nature of the available sub-surface observations (with $\sim 10$ vertical profiles per day). Note that the unusually dense coverage of T/S profiles in the English Channel during this month was due to an Ifremer research cruise.

These observations were passed through our observation operator to extract the model equivalent value from the Nature Run at the correct position and at the nearest model time-step to the observation time. To make these perfect 'observations' of the Nature Run's truth realistic, measurement and representation error must be added. For both SST and T/S profile observations, we added random Gaussian noise to each observation with platform-specific errors. SST measurement errors are defined according to the errors supplied with the observations with values from 0.3–0.6 K depending on the platform. T/S measurement error standard deviations are specified as follows: XBTs 0.15 K; Argo 0.002 K, 0.01 PSU; CTDs 0.002 K, 0.005 PSU; moored buoys 0.05 K, 0.02 PSU; and other types/unknown 0.1 K, 0.05 PSU (taken from Ingleby and Huddleston (2007), https://www.pmel.noaa.gov/gtmba/sensor-specifications, and from http://www.argo.ucsd.edu/FAQ.html). In addition to random measurement errors, to include some simulated representation error in the observations we have added a perturbation

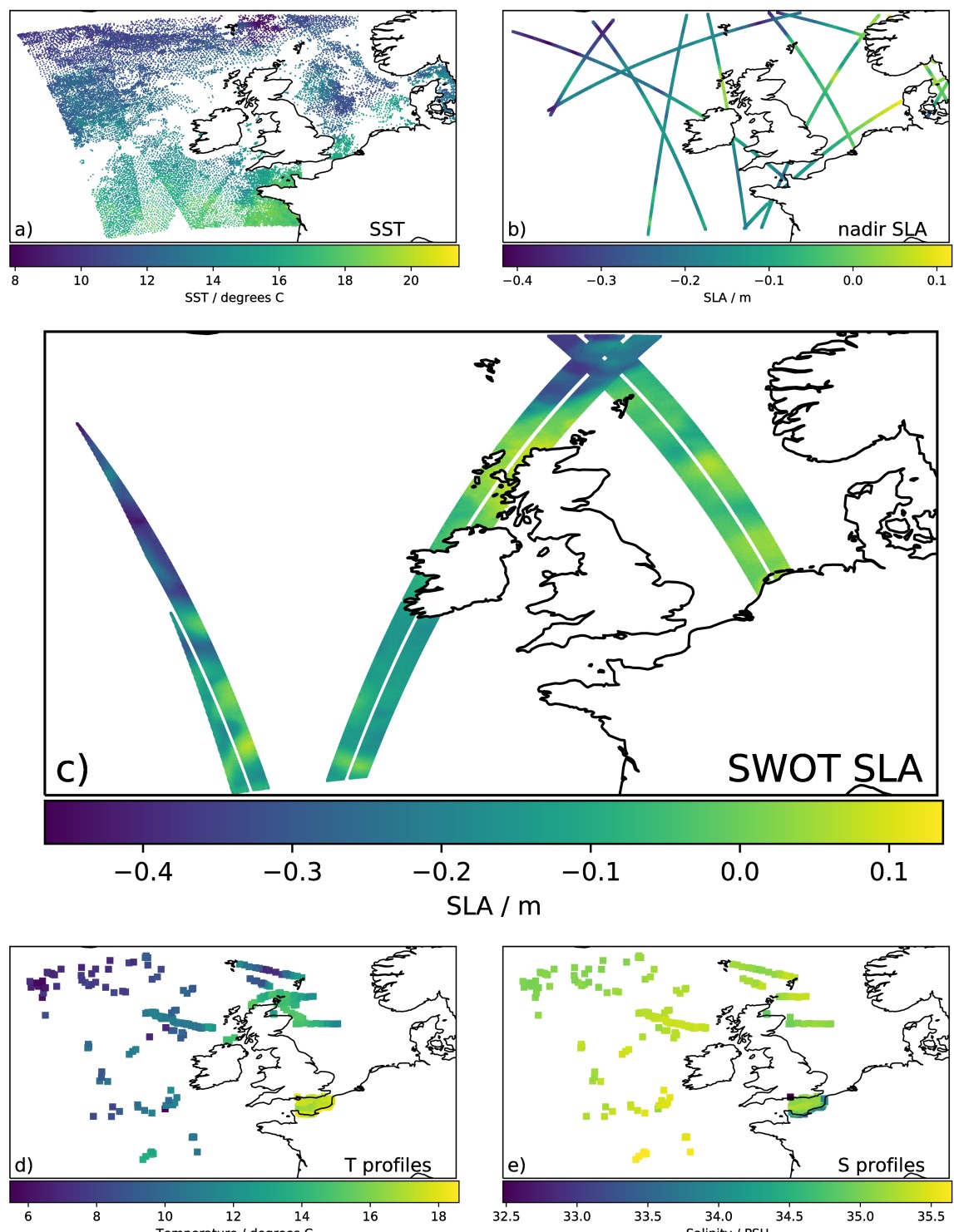

**Figure 5.** Example observation coverage for 1-day (31 Oct 2017) for a) SST, b) nadir SLA, and c) SWOT SLA observations, and 1-month (Oct 2017) for d) temperature and e) salinity profile observations.

to the location of each observation (as sampled from the Nature Run gridded fields). This is done by perturbing the real latitude
and longitude of each observation by a random amount sampled from a Gaussian distribution with standard deviation of 0.03°.

**Table 2.** SWOT swath and all nadir error statistics (RMSE and extrema, cm) for 1-month (July 2017) of simulated observations. Each component of the simulated errors is given for our experimental set-ups with the full swath (SWOT), half-swath (HalfSWOT), and the uncorrelated-error experiment (LowErrSWOT).

| Error | RMSE (cm) | Extr. | RMSE (cm) | Extr. | RMSE (cm) | Extr. |
|---|---|---|---|---|---|---|
| | **SWOT swath** | | | | | |
| | SWOT | | HalfSWOT | | LowErrSWOT | |
| All | 6.2 | 39 | 4.2 | 24 | 1.2 | 7 |
| Phase | 4.9 | 26 | 3.0 | 15 | - | - |
| Roll | 3.1 | 16 | 1.9 | 9 | - | - |
| Timing | 1.8 | 6 | 1.8 | 6 | - | - |
| KaRIn | 1.2 | 7 | 1.2 | 7 | 1.2 | 7 |
| Baseline Dilation | 0.6 | 4 | 0.2 | 1 | - | - |
| Residual Path Delay | 0.5 | 3 | 0.5 | 2 | - | - |
| | **All nadir** | | | | | |
| All | 1.4 | 6 | 1.4 | 6 | 1.4 | 6 |

## 4.2 Altimeter observations

To simulate the nadir and swath altimeter measurements of SLA, we have used the SWOTsimulator of Gaultier et al. (2016). This tool uses user-supplied model SSH fields and produces SWOT-like swath and nadir observations with random realisations of instrumental and geophysical errors. The SWOTsimulator was also used to simulate along-track nadir observations from Jason-3, Sentinal-3A, Sentinal-3B, and AltiKa to emulate the existing altimeter constellation: each of these observations has the same along-track resolution and spatial and temporal sampling as the real observations. An example of the spatial coverage of daily SLA from the existing nadir altimeters and from SWOT is shown in Figure 5, where there are on average $\sim 3,000$ nadir SLA observations and $\sim 100,000$ swath altimeter SLA observations across this domain each day. This demonstrates the 2D nature of SWOT SLA observations in comparison to the current network of nadir altimeters (note that the point-size of the nadir SLA observations is larger than the actual cross-track resolution to allow the observations to be seen on this scale), but also highlights the significant gaps between tracks on daily time-scales caused by the long repeat cycle of the SWOT satellite (21 days). Future constellations of swath altimeters could address this mismatch in the spatial and temporal sampling.

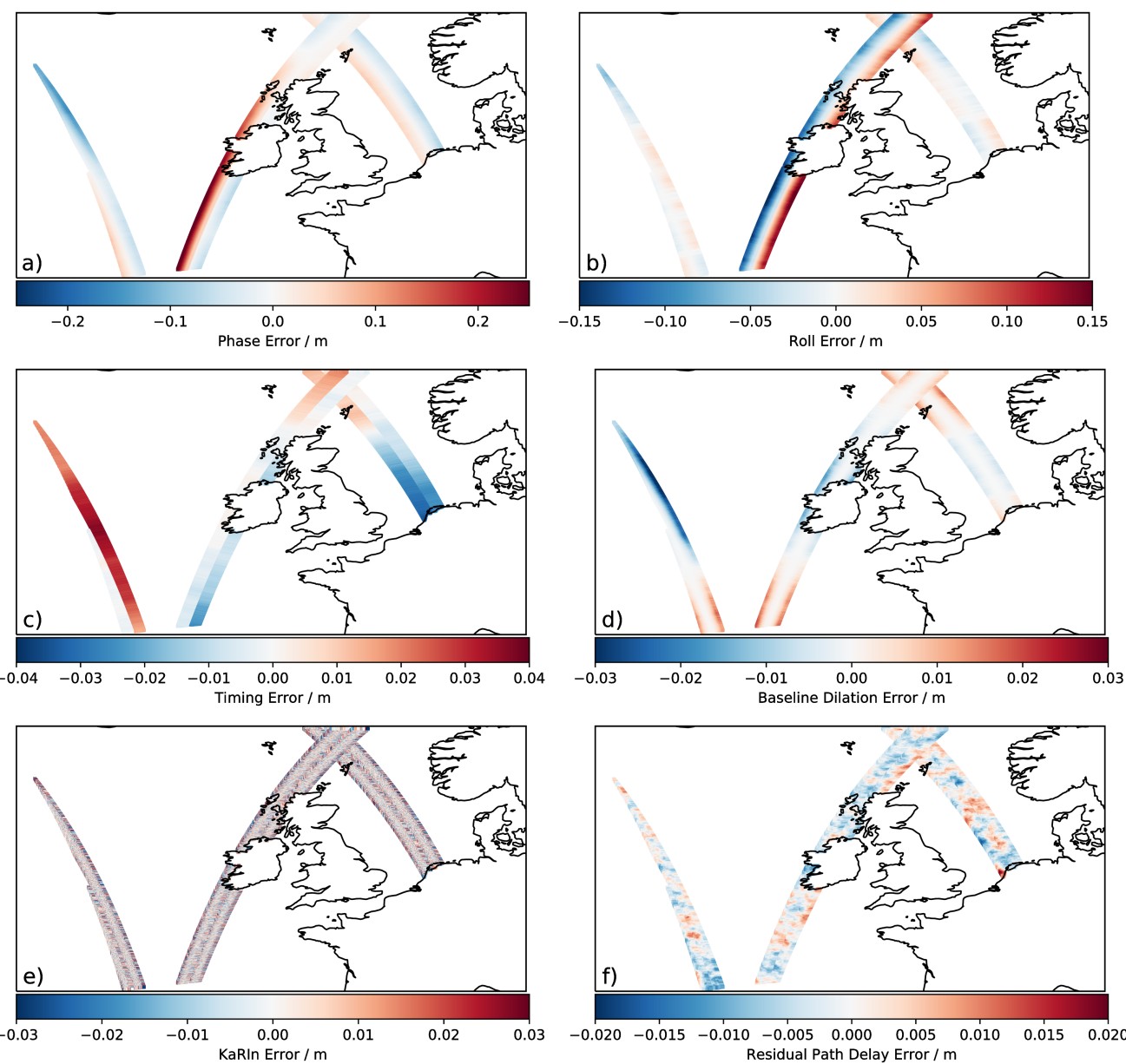

**Figure 6.** Individual components of the SWOT errors for an example day (31 Oct 2017): a) phase error, b) roll error, c) timing error, d) baseline dilation error, e) KaRIn error and f) residual path delay error. Note the difference in the scales for each error component.

To produce the simulated observations, we supplied the SWOTsimulator with 25-hour mean fields from the Nature Run to approximately remove the effect of tides. SWOT observations are expected to be provided on a 1–2 km grid (Morrow et al., 2018), so the default SWOTsimulator settings were used to produce the simulated SWOT observations on a 2x2 km grid within the swath. Since we are using the same model for the Nature Run and OSSEs, the mean dynamic topography (MDT) is known perfectly which introduces a necessary idealisation in our experiments. While the real MDT is expected to be reasonably well specified off-shelf, the sparseness of observations limits its accuracy on-shelf.

**Table 3.** SSH, surface current, and total column temperature and salinity RMSE (calculated at all model grid-points) for the on- and off-shelf regions from each experiment. The percentage change in RMSE compared to the Control Run is shown in parentheses.

| | SSH (m) | | Temperature (K) | | Salinity (PSU) | | Surface Current Speed (m/s) | |
|---|---|---|---|---|---|---|---|---|
| **Off-shelf** | | | | | | | | |
| Free Run | 0.061 | (+79%) | 0.510 | (+57%) | 0.069 | (+30%) | 0.120 | (+15%) |
| Control | 0.034 | | 0.324 | | 0.053 | | 0.104 | |
| HalfSWOT | 0.037 | (+9%) | 0.352 | (+9%) | 0.052 | (-2%) | 0.111 | (+7%) |
| SWOT | 0.041 | (+21%) | 0.391 | (+21%) | 0.054 | (+2%) | 0.114 | (+10%) |
| LowErrSWOT | 0.027 | (-21%) | 0.340 | (+5%) | 0.051 | (-4%) | 0.091 | (-13%) |
| HalfSWOT_5km | 0.032 | (-6%) | 0.324 | (0%) | 0.052 | (-2%) | 0.104 | ( 0%) |
| HalfSWOT_20km | 0.036 | (+6%) | 0.328 | (+1%) | 0.053 | ( 0%) | 0.106 | (+2%) |
| SWOT_5km | 0.036 | (+6%) | 0.334 | (+3%) | 0.052 | (-2%) | 0.106 | (+2%) |
| SWOT_20km | 0.037 | (+9%) | 0.326 | (+1%) | 0.053 | ( 0%) | 0.107 | (+3%) |
| **On-shelf** | | | | | | | | |
| Free Run | 0.021 | (+24%) | 0.612 | (+90%) | 0.137 | (+20%) | 0.035 | (+3%) |
| Control | 0.017 | | 0.322 | | 0.114 | | 0.034 | |
| HalfSWOT | 0.018 | (+6%) | 0.328 | (+2%) | 0.108 | (-5%) | 0.036 | (+6%) |
| SWOT | 0.021 | (+24%) | 0.366 | (+14%) | 0.115 | (+1%) | 0.039 | (+15%) |
| LowErrSWOT | 0.017 | (0%) | 0.314 | (-2%) | 0.108 | (-5%) | 0.034 | ( 0%) |
| HalfSWOT_5km | 0.017 | (0%) | 0.325 | (+1%) | 0.111 | (-3%) | 0.035 | (+3%) |
| HalfSWOT_20km | 0.017 | (0%) | 0.324 | (+1%) | 0.113 | (-1%) | 0.035 | (+3%) |
| SWOT_5km | 0.018 | (+6%) | 0.333 | (+3%) | 0.110 | (-4%) | 0.036 | (+6%) |
| SWOT_20km | 0.017 | ( 0%) | 0.325 | (+1%) | 0.112 | (-2%) | 0.035 | (+3%) |

The wide-swath altimeter observations are subject to a number of sources of error, including phase error, roll error, timing error, KaRIn noise, baseline dilation error, and residual path delay error (see Gaultier et al., 2016). These errors are significantly

larger than that associated with current nadir SLA observations. Each of the noise components is shown in Figure 6 for an example day and the standard deviation and extreme values from 1-month of simulated observations are summarised in Table 2.

The phase error results from changes in the relative phase between two signal paths in the interferometer which affects each side of the swath independently. The baseline dilation error also relates to distortions in the antennas, while the roll error is an estimate of the error in the satellite roll angle. The timing error is caused by a drift in the timing of the instrument signal propagation and is constant across-track. The residual path delay error is the estimate of the error resulting from uncertainties in the radar retrieval due to atmospheric water vapour. Finally, the Ka-band Radar Interferometer (KaRIn) error is an uncorrelated error which varies with sea-state and the across-track distance from the nadir point. For these experiments we have used the default 2 m significant wave height in the simulation of the SWOT observation errors. In regions with large waves, such as the Southern Ocean, the KaRIn errors will limit the accuracy of SWOT observations. The phase and roll errors in particular can introduce spatially correlated errors in excess of 10 cm along or across a swath. As shown by Figure 6, the length-scale of these correlations can also be at least as large as our model domain.

To test the limitations imposed by these large correlated errors on the assimilation of SWOT observations, we have run the OSSEs detailed in Table 1 and so have created simulated SWOT observations for each experiment: the full swath with all error components (for the SWOT runs); the inner half of the swath with all error components (for the HalfSWOT runs); and the full swath with only the uncorrelated KaRIn errors (for the LowErrSWOT run). The effect of these limitations on the total observation error (and on each component of the error) is shown in Table 2. Reducing the width of the swath by half results in a drop of >30% in the RMSE (root mean square error) and removes the most extreme values. The LowErrSWOT observations include only the KaRIn errors and have an RMSE closer to that of the current nadir altimeter observations. In all of the experiments the SLA observations have been quality-controlled before assimilation. The observation-model background differences and the observation and background errors are used to assess the probability of gross error. Those observations with a probability of gross error greater than 50% are not assimilated. This however only removes ∼0.3% of the SLA observations in the SWOT experiment. More stringent QC could remove more of the observations which are affected by large correlated errors, but at the expense of valid observations where the model most needs to be constrained.

## 5  Results

In this section we will detail the impact of assimilating the simulated SWOT observations by comparing each experiment with the truth provided by the full 4D fields from the Nature Run. The gridpoint-by-gridpoint differences between each OSSE and the Nature Run allow us to examine the impact of the assimilation over the full domain. This means we are not hindered by incomplete observation sampling as would be the case with a system assimilating real observations. Since the SSH signal is dominated by barotropic processes in shallow water and baroclinic processes in deeper water, we present these comparisons separately for the on-shelf and off-shelf regions as marked in Figure 1. The impact of each experiment on the SSH, surface current speed, temperature and salinity RMSE calculated at all model grid-points is summarised in Table 3.

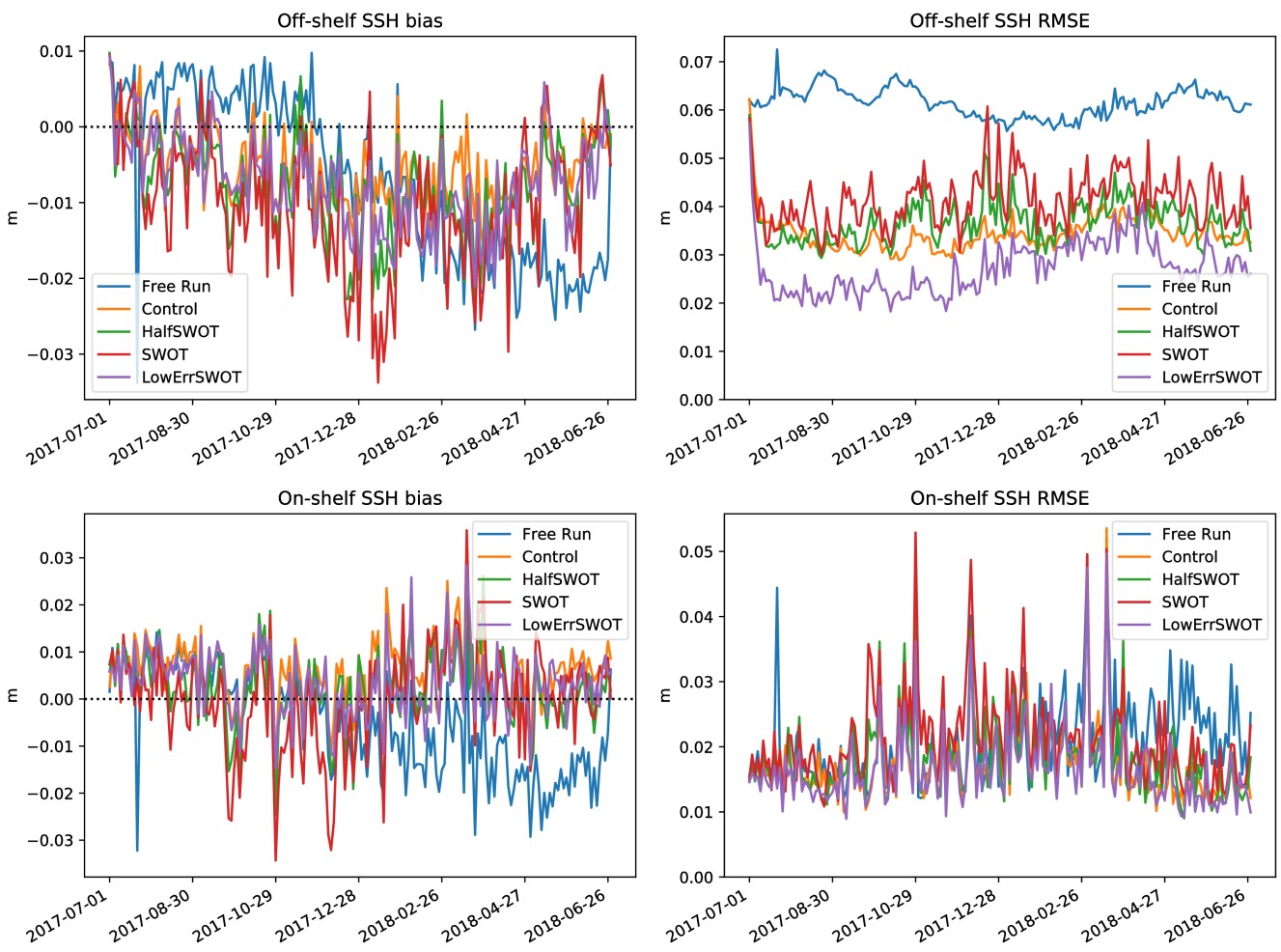

**Figure 7.** SSH bias (m, left) and RMSE (right) for the off-shelf (top) and on-shelf (bottom) regions.

## 5.1 Impact on SSH

The impact of assimilation is clearly positive in the off-shelf SSH RMSE in all experiments, each of which reach a stable
RMSE within the first month of the runs (see Figure 7), but with considerable seasonal variation. However, the Control is
superior to the HalfSWOT and SWOT experiments which assimilate the SWOT observations with the full correlated errors.
The HalfSWOT and SWOT experiments increase the off-shelf SSH RMSE (calculated over the full run excluding the first
month, see Table 3) by 9% and 21% respectively relative to the Control, while the LowErrSWOT experiment which did not
include the large correlated errors shows a reduction of 21% in the RMSE relative to the Control. The impact on the off-shelf
bias is less clear. When the bias in the mixed layer depth increases into winter the Free Run SSH bias increases. However,

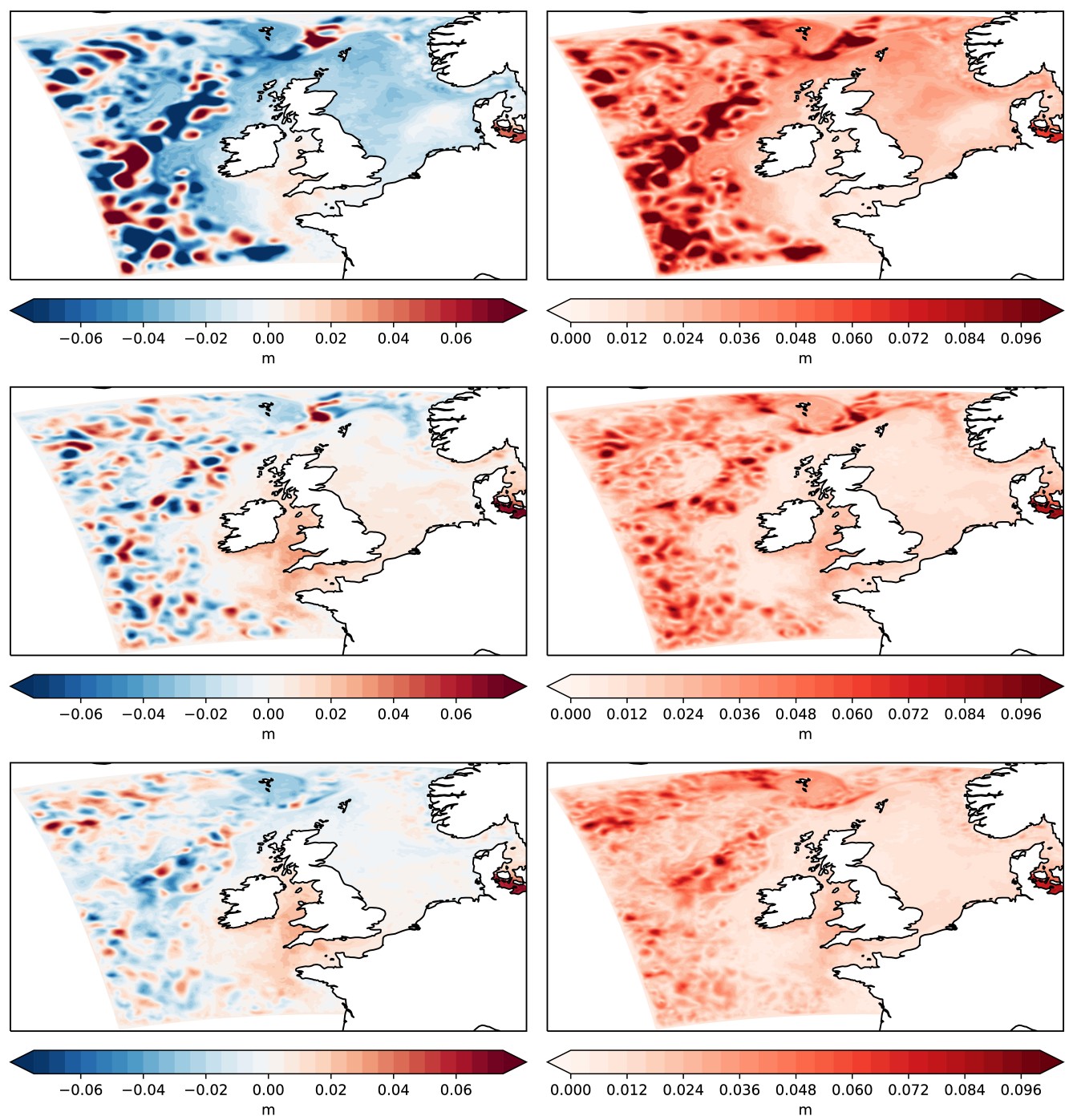

**Figure 8.** June 2018 average SSH bias (left) and RMSE (right) for Free Run (top), Control (middle), LowErrSWOT experiments (bottom).

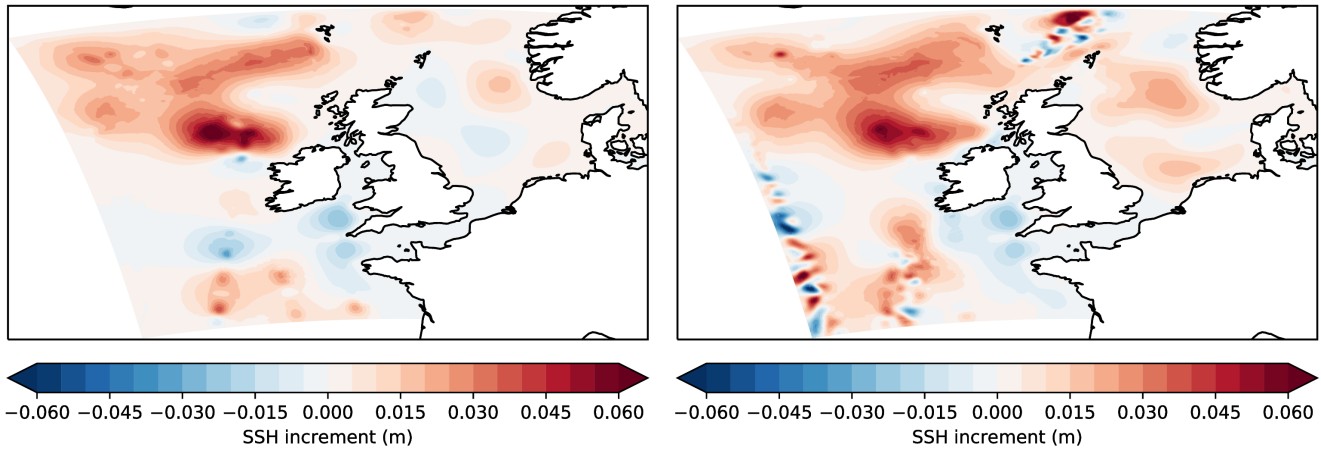

**Figure 9.** SSH increments for an example day (31 Oct 2017) from the Control (left) and LowErrSWOT experiments (right).

all assimilative runs show similar slightly negative biases, suggesting that the bias is relatively well-constrained by profile assimilation.

The modelled SSH variability on-shelf is lower than in the off-shelf regions. However, as with the off-shelf region, the Free Run shows a drift in the on-shelf SSH bias, which is well-constrained in all of the assimilative runs. In absolute terms, there is little difference in the on-shelf SSH RMSE between the assimilative experiments. However, in percentage terms the impact in the HalfSWOT and SWOT experiments is similar to that seen off-shelf, with an increase in the on-shelf SSH RMSE (calculated over the full run excluding the first month) of 6% and 24% respectively. However, the LowErrSWOT experiment shows a negligible difference in the RMSE relative to the Control.

One notable feature is the frequent large spikes in the on-shelf SSH RMSE, which are present in the Free Run and the assimilative runs. These appear relatively unaffected by the assimilation of the standard network of observations, or by the additional SWOT observations. These are likely due to errors in the surface forcing (i.e., differences between the ECMWF IFS and MetUM surface fluxes) which can have a large impact on the SSH during surge events. This reflects similar RMSE spikes seen in our operational system and is an issue we aim to address in the near future. However, the experimental set-up here where the SLA observations were simulated using 25-hour model mean fields is sub-optimal for constraining these short period events. To a lesser extent, high frequency RMSE increases are also seen in the off-shelf RMSE time-series. This may be due to the misplacement of eddies; due to the distance and time between SWOT swaths, much of the mesoscale structure will still be unobserved. Notably, the experiments which include correlated errors in the SWOT observations show a marked increase in the high frequency RMSE variability.

Spatial maps of the mean bias and RMSE over the final month of the experiments (June 2018) further demonstrate the improvement both on- and off-shelf when assimilating SWOT observations without correlated errors. Figure 8 shows the progression of the bias and RMSE from the Free Run to the Control and then to the LowErrSWOT experiment. The large SSH

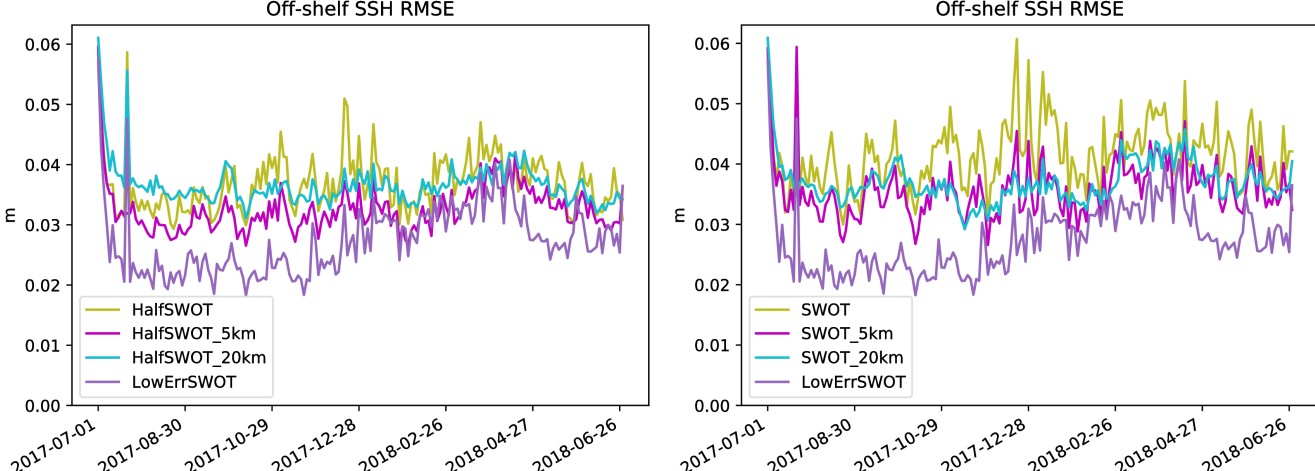

**Figure 10.** Off-shelf SSH RMSE for the HalfSWOT (left) and SWOT (right) experiments with the corresponding experiments where SWOT observations have been averaged over 5 km and 20 km radii. The LowErrSWOT experiment is also show in both cases.

bias across the domain and the increased RMSE off-shelf seen in the Free Run are clearly improved when assimilating the standard network of observations, with a significant further improvement when assimilating SWOT observations without correlated errors. Although there is a change in the sign of the bias in the Irish and Celtic Seas from the Free Run to the Control, the assimilation of uncorrelated SWOT observations improves the bias and RMSE.

One of the expected benefits of assimilating wide-swath altimeter observations from SWOT is the ability to better constrain small-scale structures which cannot be fully-resolved by along-track altimeter observations. Figure 9 shows SSH increments from our assimilation scheme for the Control and LowErrSWOT experiments from a single day, corresponding to the observation coverage shown earlier in Figure 5. This demonstrates that the assimilation of SWOT observations (in the LowErrSWOT experiment) attempts to add small-scale structures which are not present in the Control experiment increments (where no SWOT observations are assimilated). However, this small-scale structure is limited by the relative sparseness of the SWOT swaths across the domain, due to the long repeat cycles and distance between swaths. In this implementation, SLA innovations were calculated from 24hour 50m model fields to remove the tidal signal from the innovations. As a result, the highest frequency differences, such as expected on-shelf, will be smoothed. However, small-scale increments are added in the off-shelf regions when assimilating SWOT.

### 5.1.1 Impact of Observation Averaging

Assimilating SWOT observations without correlated errors can clearly have a large positive impact. By limiting the SWOT data to the inner half of the swath, we have shown that some of the degradation caused by correlated errors can be reduced. We next considered whether observation-averaging within the SWOT swath could further reduce the problems associated with assimilating SWOT observations with correlated errors. Averaging radii of 5 km and 20 km were applied to SWOT data from

both the full and half swath widths. We chose these radii to test a minimal level of averaging (5 km) and a larger radius similar to that used in other studies (20 km, e.g., Carrier et al., 2016). Although the length-scales of the correlations apparent in the simulated SWOT observations can be considerably longer than 20 km (see Figure 6), we chose not to test larger averaging radii as this would severely degrade the unique ability of SWOT observations to resolve the surface topography of the ocean.

The observation errors used in the assimilation scheme were not changed in the experiments where SWOT observations were median-averaged. Although it may be beneficial to change the observation errors depending on the chosen level of averaging, the main aim of the averaging was to reduce the effect of the largest correlated errors rather than reducing the random component of the errors. A more detailed examination of the impact of the observation errors will be carried out in the future.

The impact of observation averaging on the SSH RMSE is shown in Figure 10. Both levels of observation averaging improve the results compared to no averaging, for the full and half swath width cases. In the experiments assimilating the full swath width, it is clear that the lower the level of SWOT observation averaging the greater is the day-to-day variability in the SSH RMSE. This is presumed to be caused by the assimilation of observations with the largest errors. Overall the best results when assimilating SWOT observations with correlated errors are found when using half the SWOT swath and 5 km observation 380    averaging (HalfSWOT_5km).

## 5.2    Impact on Sub-surface Temperature and Salinity

### 5.2.1    Off-shelf T/S

In the off-shelf region, the near-surface temperature bias found in the Free Run (see Figure 3) is very quickly removed and the RMSE significantly reduced in all assimilative runs. This is not surprising given the large quantity of SST observations 385    assimilated in all OSSE runs. Maps of the vertically-averaged temperature bias and RMSE for the final month of the simulations demonstrate the spatial differences between the experiments (Figure 11). Here the improvement between the Free Run and Control is dominated by the improvement within the mixed layer from SST assimilation.

Although these maps show a small degradation in the bias and RMSE when assimilating SWOT observations without correlated errors (see bottom row of Figure 11), the average vertical profile of the off-shelf temperature bias and RMSE for 390    all experiments over the full year (shown in Figure ??) shows a marginal additional reduction in the near-surface temperature RMSE in the LowErrSWOT experiment. The assimilation of the standard network of observations appears to introduce a sub-surface bias around 30–80 m depth in the Control run, which is exacerbated in the SWOT, HalfSWOT and LowErrSWOT experiments, explaining the overall degradation seen in the vertically-averaged statistics. This effect, due to the method used to spread SST information in the vertical, was noted in King et al. (2018) and methods to avoid or reduce it are currently being 395    investigated. It is known that profile observations can constrain this increase in the thermocline bias and RMSE, but additional altimeter observations appear to exacerbate the issue.

Below $\sim 500$ m, the LowErrSWOT experiment again shows a marginal improvement in RMSE compared to the Control, while the HalfSWOT experiment shows little change and the SWOT experiment a degradation. Overall, apart from the known

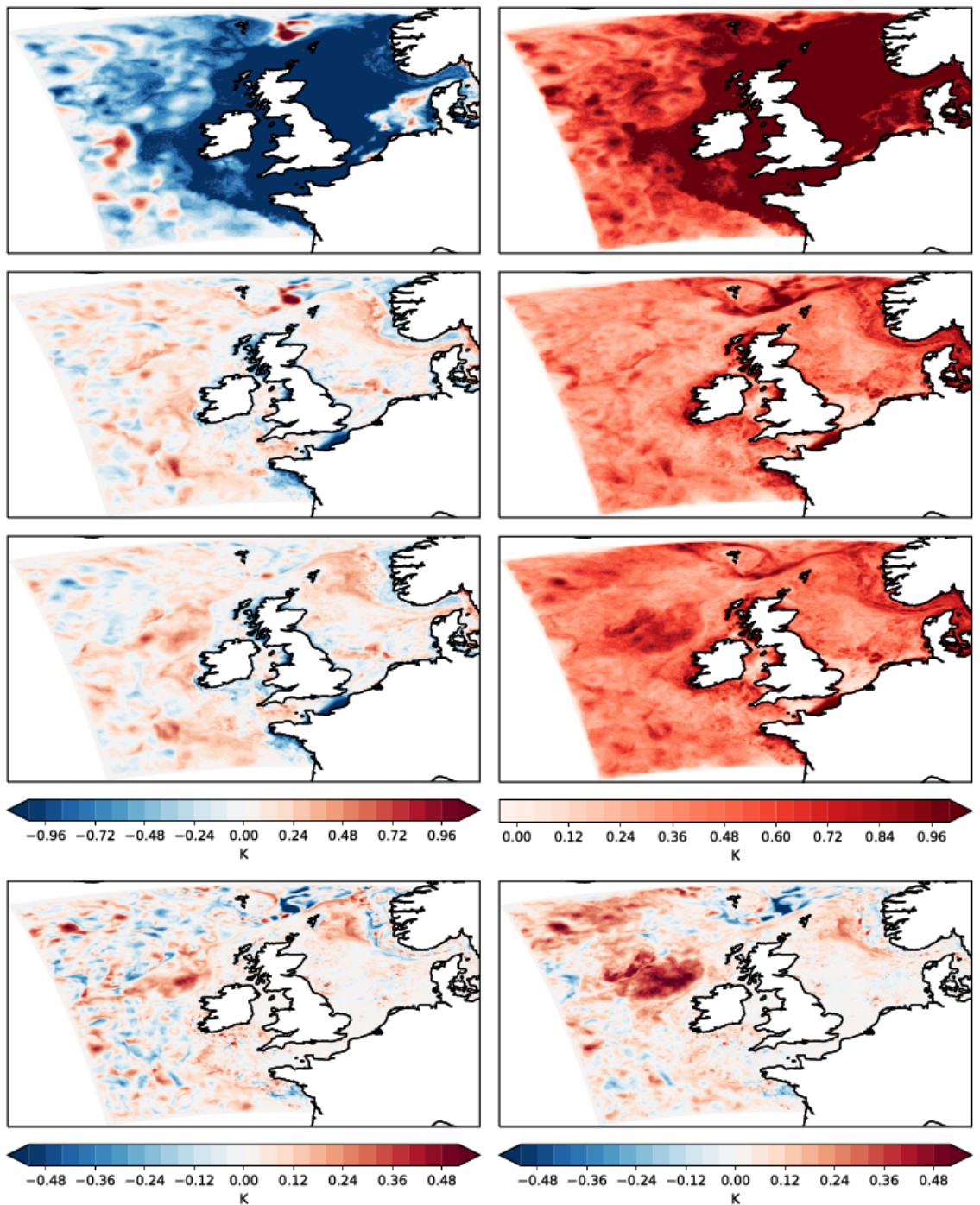

**Figure 11.** June 2018 vertically-averaged temperature bias (left) and RMSE (right) for Free Run (top), Control (second row) and LowErr-SWOT experiments (third row). The difference (Control-LowErrSWOT) in the absolute bias and RMSE between the Control and LowErr-SWOT experiments is also shown (bottom row) to highlight the impact of the additional SWOT observations.

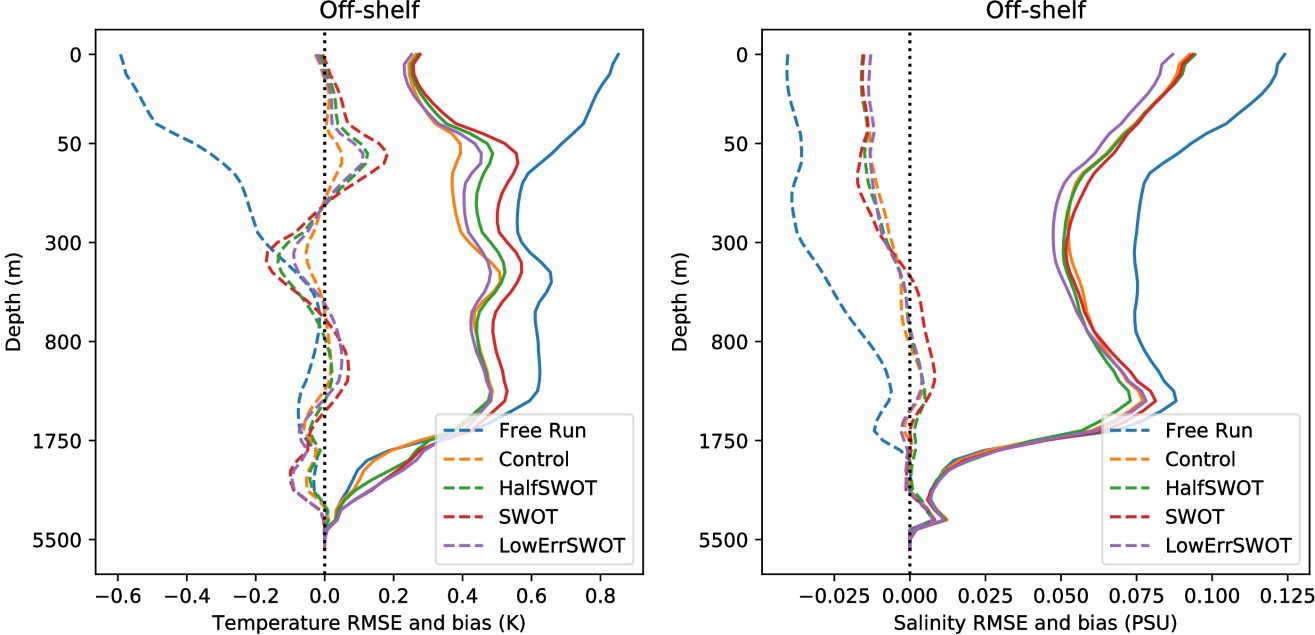

**Figure 12.** Off-shelf temperature and salinity RMSE (solid) and bias (dashed) averaged over the full year of the experiments. Note the log-scale used for depth to highlight the differences in the upper ocean.

issues near the thermocline, the additional SWOT observations can reduce the off-shelf temperature errors, but the full corre-
lated SWOT errors lead to a degradation at almost all depths.

Off-shelf salinity biases are also rapidly reduced by assimilation in the Control run, with a quicker decrease in bias seen in the LowErrSWOT experiment (not shown). Near-surface salinity RMSE is also reduced in the Control relative to the Free Run, but not to the extent as for temperature due to the relative sparseness of the in situ salinity observations. The salinity is not affected by the known issue discussed above where SST assimilation leads to inflated biases near the thermocline.

Maps of the vertically-averaged salinity bias and RMSE for the final month of the simulations (Figure 13) show a small improvement across the off-shelf region when assimilating uncorrelated SWOT observations. The average off-shelf salinity bias and RMSE over the full year of the experiments shown in Figure **??** further demonstrates that all assimilative runs have improved salinity errors compared to the Free Run. Overall, the additional SWOT observations in the LowErrSWOT experiment decrease the salinity biases and RMSE at all depths, but assimilating SWOT observations with the full correlated errors
leads to a slight degradation compared to the Control, especially when assimilating the full swath.

### 5.2.2   On-shelf T/S

As discussed in Section 3.2, our assimilation scheme applies unbalanced changes to SSH along with balancing (baroclinic) changes to the vertical temperature and salinity. These balancing changes are only applied where the water column is reasonably

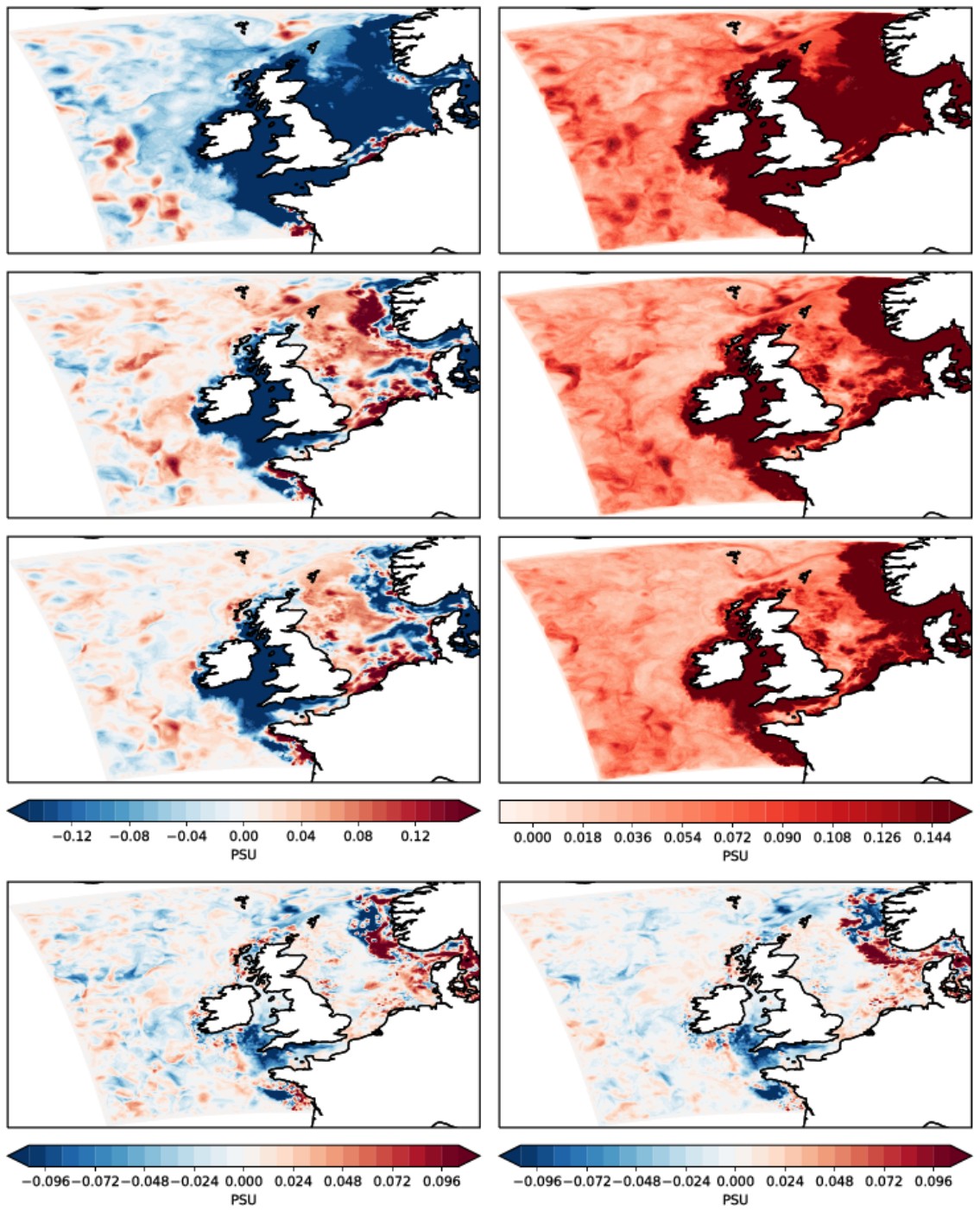

**Figure 13.** June 2018 vertically-averaged salinity bias (left) and RMSE (right) for Free Run (top), Control (second row) and LowErrSWOT experiments (third row). The difference (Control-LowErrSWOT) in the absolute bias and RMSE between the Control and LowErrSWOT experiments is also shown (bottom row) to highlight the impact of the additional SWOT observations.

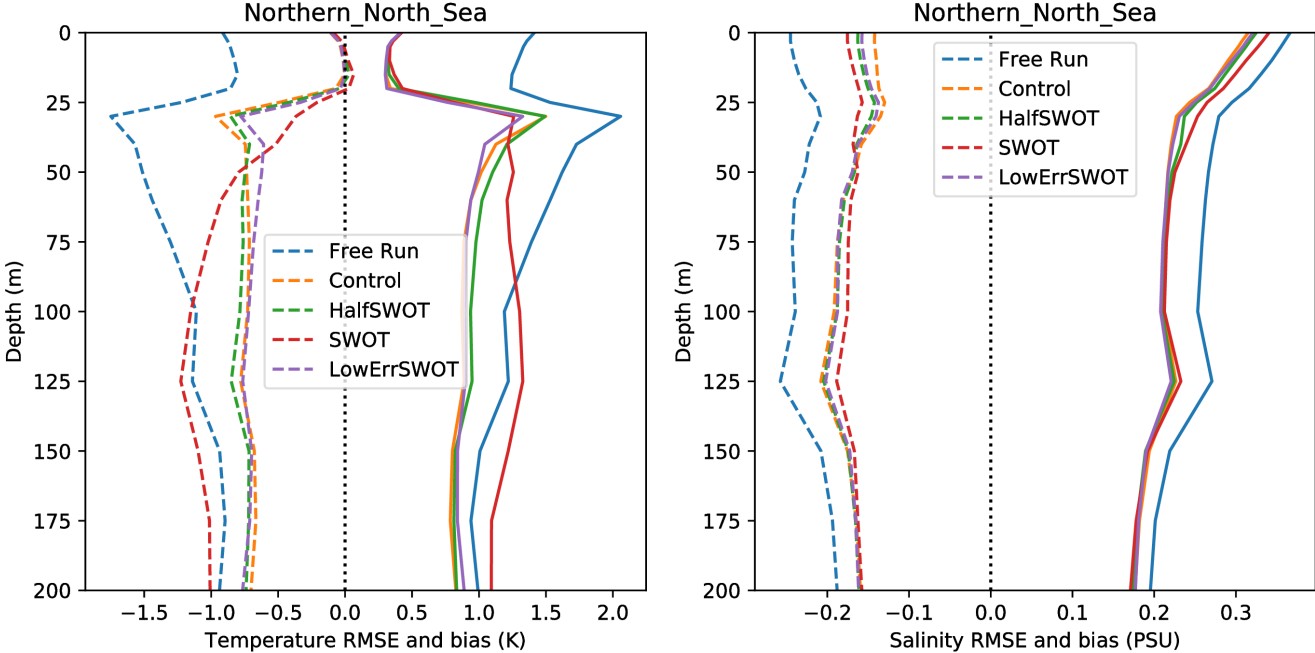

**Figure 14.** Northern North Sea temperature and salinity RMSE (solid) and bias (dashed) averaged over June–October when the full tempeature and salinity balanced changes due to SSH assimilation are applied.

well-stratified (as demonstrated in Figure 4). We define this as when the top-to-bottom temperature difference exceeds 2°C. As
a result of this, in regions/periods where no balanced changes to temperature and salinity are applied when assimilating SSH, there is no appreciable difference between the temperature and salinity in the various assimilating experiments. That is, while the Control is superior to the Free Run in terms of the temperature and salinity bias and RMSE, no further improvement is gained from assimilating additional SLA observations. Therefore for the on-shelf region, we have focussed our assessment on the Northern part of the North Sea during the period June–October when the full SSH balance is applied.

As for the off-shelf region, the near-surface on-shelf temperature bias and RMSE are constrained by the assimilation of the available SST observations (see Figure 11). In particular, in the well-mixed regions in the Southern North Sea the full temperature profile is well-constrained by SST assimilation and there is no additional benefit from assimilating additional SLA observations. This is also clear in the vertical profile statistics for the Northern North Sea in Figure 14 where in the upper 25 m there is almost no difference in the on-shelf temperature RMSE or bias between all of the assimilative runs. However, the
average temperature bias and RMSE for the Northern North Sea in Figure 14 show a small improvement in the temperature errors between 25 m and 100 m for the LowErrSWOT experiment in this region and period where the full SSH balance is applied. On the other hand, the SWOT experiment with correlated errors shows significantly larger temperature RMSE below ∼50 m, though this RMSE rise is limited when assimilating only the inner half of the swath in the HalfSWOT experiment.

In contrast, while the on-shelf salinity bias and RMSE are significantly improved by the assimilation of the standard network of observations, they are little changed over much of the on-shelf region when assimilating the additional SWOT observations (Figure 13) and large biases remain where in situ measurements of salinity are sparse. However, there is a significant reduction in the bias and RMSE in the Celtic Sea when assimilating SWOT observations, and a mixed impact along the Norwegian Trench. The average salinity bias and RMSE for the Northern North Sea in Figure 14 is marginally degraded in the upper 25 m by the assimilation of SWOT observations even in the HalfSWOT and LowErrSWOT experiments, likely due to the differences seen along the Norwegian Trench. Also, while at some depths there is an improvement in the salinity bias in the SWOT experiment, this is accompanied by a small increase in RMSE below ∼50 m.

In summary, on-shelf temperature and salinity errors can be improved by the assimilation of SWOT observations without correlated errors. Restricting the SWOT data with the full correlated errors to the inner half of the swath limits the negative impacts. However, when the full swath width SWOT observations contain realistic correlated errors, there is a degradation in both temperature and salinity at almost all depths.

### 5.2.3   Impact of Observation Averaging

In general, observation averaging was found to improve the off-shelf temperature bias and RMSE in experiments with both the full and half-swath width (compared to no observation averaging, see Table 3). Near the thermocline, where there is a known issue caused by the SST assimilation (discussed above), the larger averaging radius leads to a lower bias and RMSE due to the lower quantity of observations assimilated (as noted above, the effect is exacerbated by additional observations). However, elsewhere the 5 km averaging radius is marginally superior. The on-shelf temperature bias and RMSE when assimilating SWOT with correlated errors are also improved by observation averaging, but in this case the 20 km radius is superior.

For salinity, there is a smaller overall impact from observation averaging. Off-shelf, while the 5 km averaging is marginally better than no averaging, using the 20 km averaging radius gives mixed results with small degradations at some depths. On-shelf, when using the full swath observation averaging improves the RMSE and bias, with the 5 km radius giving best results, but when using the half-swath observation averaging degrades the results, with the larger radius showing the largest degradation. Overall, a 5 km observation averaging radius appears to give most improvements to the temperature and salinity statistics across the domain with the fewest and smallest degradations.

### 5.3   Impact on surface currents

A comparison of daily and monthly mean surface current speeds (not shown) highlights that the major surface current features of the Nature Run are also present in the Free Run. However, with no observations to constrain the surface currents many of the eddies and current meanders are misplaced. The Control run assimilating the standard network of observations improves the positions of individual eddies and straightens the current through the the Faroe-Shetland channel. With the addition of the uncorrelated SWOT observations, the LowErrSWOT experiment further improves the simulation of the surface current features.

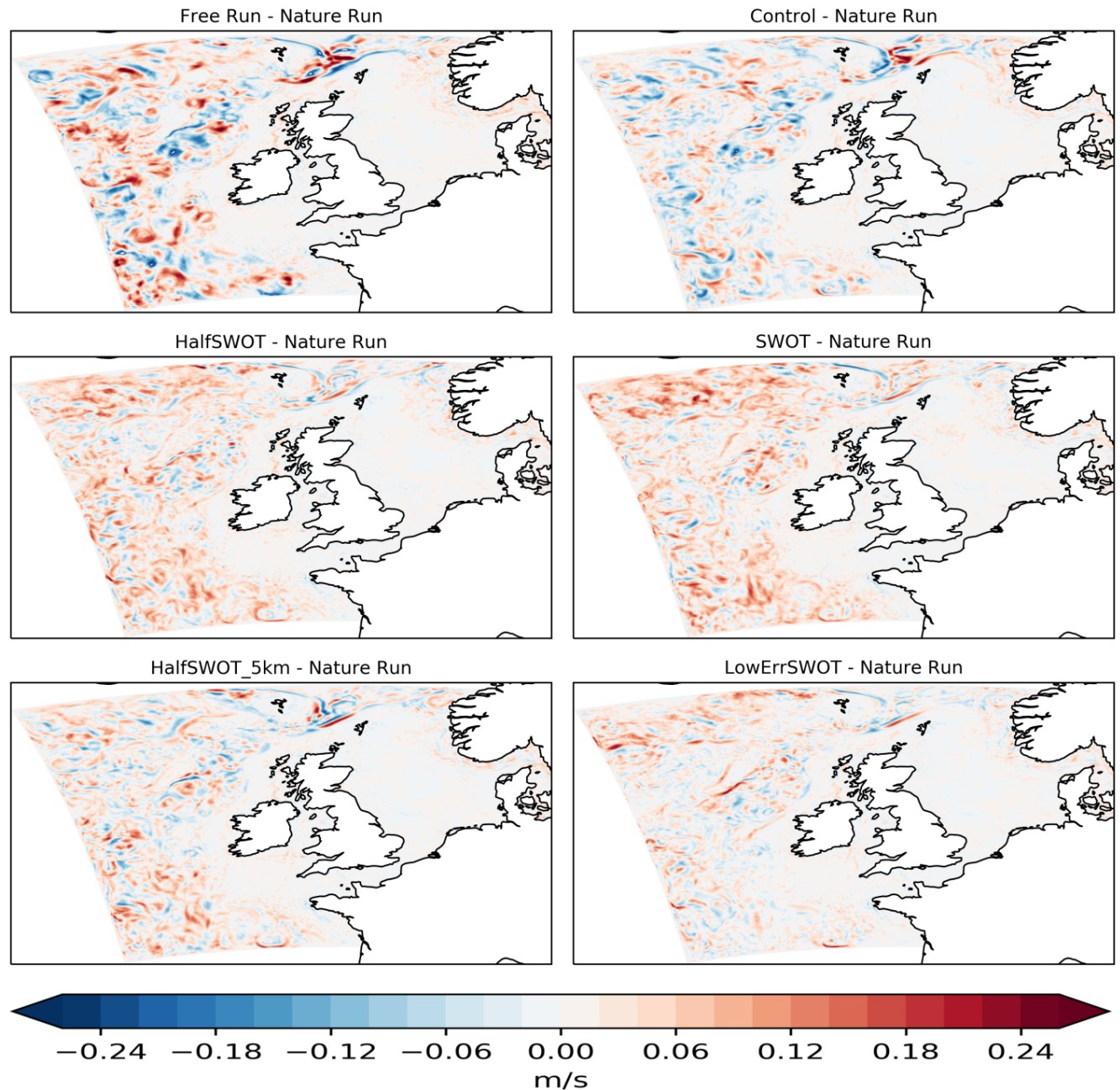

**Figure 15.** Monthly (June 2018) mean surface current speed error (compared to Nature Run) from the Free Run (top left), Control, HalfSWOT, SWOT, HalfSWOT_5km and LowErrSWOT experiments (bottom).

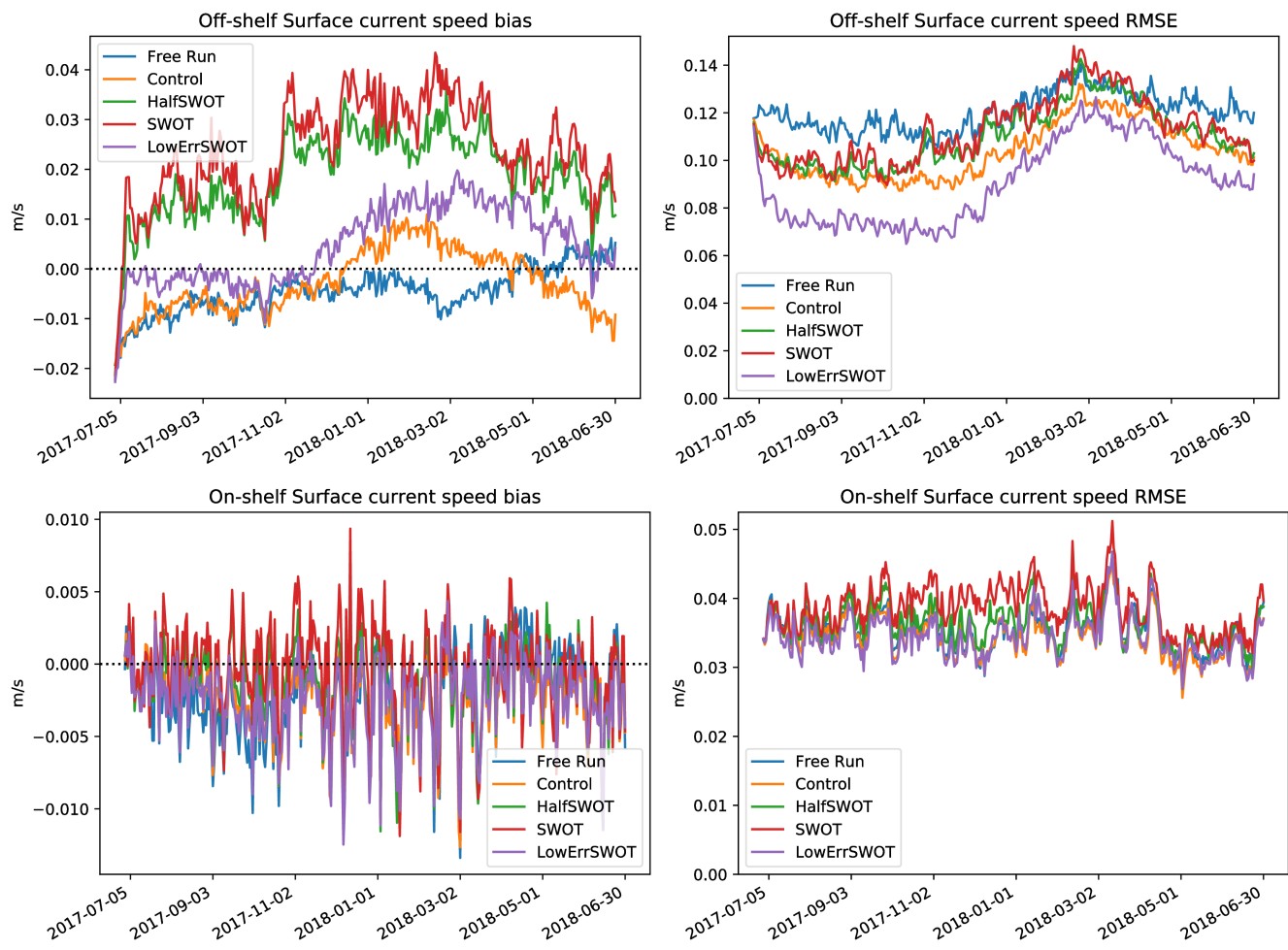

**Figure 16.** Surface current speed bias (left) and RMSE (right) for the off-shelf (top) and on-shelf (bottom) regions.

This improvement in the positioning of surface current features is clearly seen in the maps of monthly surface current errors shown in Figure 15. A reduction in errors in the surface currents is seen at large- and small-scales suggesting the mesoscale structures are better initialised when assimilating the SWOT observations without correlated errors. In particular, there is a clear additional benefit of assimilating SWOT observations with uncorrelated errors. However, when assimilating SWOT observations with correlated errors, the daily difference fields show correlated features aligned with the SWOT swaths which manifest in the monthly errors as a generally positive bias - that is, the average surface current speed is erroneously increased when assimilating SWOT observations with large correlated errors.

The area-averaged impact of assimilation in each of the experiments is summarised in the time-series of bias and RMSE for the on- and off-shelf regions in Figure 16. As with the SSH RMSE, the surface current reaches a stable RMSE within the first month of the runs. The HalfSWOT and SWOT experiments increase the off-shelf surface current speed errors (calculated over

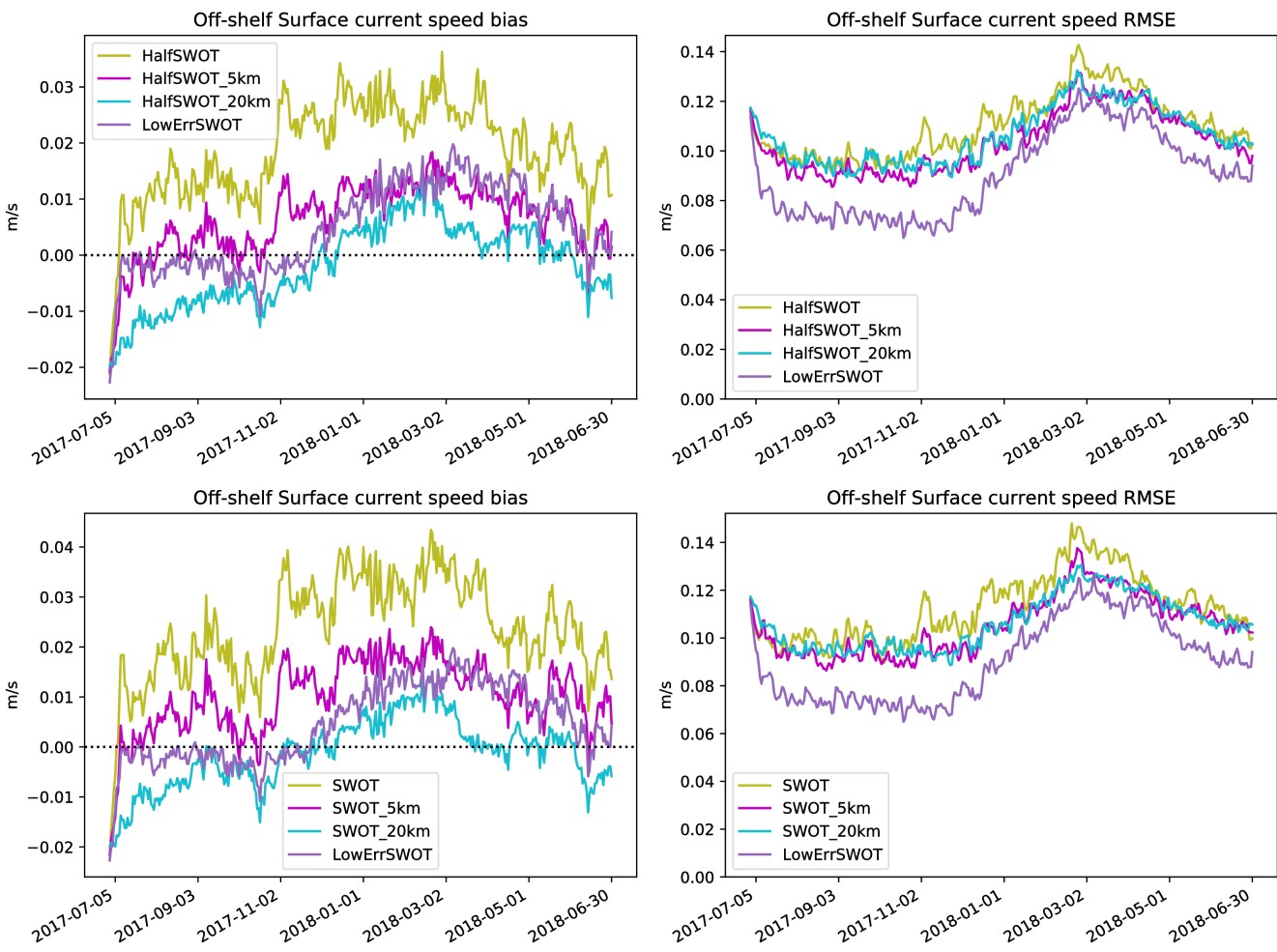

**Figure 17.** Off-shelf surface current speed bias (left) and RMSE (right) for the HalfSWOT (top) and SWOT (bottom) experiments showing the impact of observation averaging. The LowErrSWOT experiment is also show in both cases.

the full run excluding the first month, see Table 3) by 7% and 10%, respectively, while the LowErrSWOT experiment which did not include the large correlated errors shows a reduction of 13% in the RMSE relative to the Control (a similar absolute change in RMSE as from the Free Run to the Control). There is also significant seasonal variation in the error reduction achieved in the LowErrSWOT experiment, with a peak RMSE reduction of ∼20%. As with the SSH RMSE, the on-shelf differences are smaller in absolute terms, but the HalfSWOT and SWOT experiments increase the on-shelf surface current speed errors by 6% and 15%, respectively, while the LowErrSWOT experiment has a negligible impact on the surface current speed errors relative to the Control.

The effect of assimilating SWOT observations on the surface current speed bias is less straight-forward. While initially the off-shelf bias is little changed from the Free Run to the Control and is significantly reduced in the LowErrSWOT experiment, in the latter half of the experiments the bias drifts in all runs. However, both the experiments assimilating SWOT observations with correlated errors show an increase in the overall RMSE and a large increase in the bias (as seen in the earlier spatial maps of surface current errors).

Overall, the assimilation of SWOT observations can better initialise the position and strength of eddies and significantly reduce the surface current RMSE, but correlated SWOT errors lead to a bias toward faster surface currents.

### 5.3.1 Impact of Observation Averaging

While we have shown that assimilating correlated SWOT observations degrades the surface current speed RMSE and bias, when observation averaging is used we find a significant decrease in the bias and RMSE compared to the experiments without averaging. The reduced bias is apparent in the maps of monthly surface current errors shown in Figure 15 where the HalfS-WOT_5km experiment shows a clear reduction in the bias relative to the Control and HalfSWOT experiments, but still larger bias than in the LowErrSWOT experiment. Figure 17 demonstrates that the 5 km and 20 km observation averaging reduces the overall RMSE by similar amounts in the HalfSWOT and SWOT experiments, but with lower variability in the RMSE over the year when averaging over the larger radius. In experiments using the full and half-swath widths, the larger averaging radius results in a lower bias over the year. However, the trend is the same in all the experiments with correlated errors, so the same underlying issue affects all of the experiments.

## 6 Discussion

Overall, the correlated errors expected in real SWOT observations hinder the full exploitation of this step change in our ability to observe the ocean mesoscale dynamics. The off-shelf SSH, surface currents, and 3D temperature and salinity can be improved by the assimilation of SWOT observations with uncorrelated errors. However, when the full correlated errors are included in the simulated SWOT observations, the SSH RMSE increases relative to the Control, surface currents are erroneously increased and there is a significant temperature and salinity degradation at almost all depths.

The assimilation of the standard network of observations introduces a sub-surface bias near the base of the mixed layer in the Control experiment, which is exacerbated by additional SWOT observations. This was noted in King et al. (2018) and may

be due to the assimilation of SST observations in our configuration of NEMOVAR. Solutions to this problem are currently being developed.

On-shelf, the temperature below the mixed layer and salinity at some depths is improved by the assimilation of SWOT observations with uncorrelated errors, but again is degraded when using the full correlated errors, and is exacerbated when using the full swath width. Due to the lower variability on-shelf, the impact of assimilating SWOT observations on SSH and the surface currents is less clear, but follows a similar pattern: the assimilation of observations with uncorrelated errors marginally decreases errors, while assimilation of observations with correlated errors leads to a degradation. In stratified regions

SLA observations contain information on vertical T/S structure which is effectively assimilated in the deeper ocean and in the stratified regions on-shelf. However, our assimilation scheme applies a ramping of the velocity balance near coasts which may be limiting the retention of these observations on-shelf and so their impact on the SSH and velocity statistics. Additionally, our approach of using 25-hour mean fields in the simulation of the SLA observations and for the background in our innovations necessarily removes high-frequency signals which dominate on-shelf. We intend to further explore adaptions to our assimilation

scheme, including whether it would be beneficial to retain the higher-frequency signals in our innovations.

Our main aim here was to investigate the limitations posed by correlated errors on the impact of SWOT. However, we have also shown that our assimilation scheme attempts to add more small-scale structures when swath altimetry is assimilated. A consequent reduction in the errors in surface currents was found on both large- and small-scales suggesting the mesoscale structure is better initialised when assimilating SWOT observations without correlated errors.

More generally, the multivariate balances used in our assimilation scheme were originally implemented for an open ocean system. It is therefore likely that the specification of the balances are sub-optimal for on-shelf assimilation. For instance, the water mass balance used may be less appropriate on the shelf, particularly where there is significant influence of rivers. Additionally, the vertical correlation length-scales which were originally parameterised for on open ocean location could be adapted for the shelf-seas. This requires further assessment to understand the effects of each observation type and with

simplified balances to understand how they operate in practice in this regime. We are currently developing global and shelf-seas ensemble systems which will provide information we can use to better represent both errors-of-the-day and the region-specific balances. In the future, we plan to use this information to adjust the balances and/or within a hybrid data assimilation system combining ensemble information with climatological error covariances. It is hoped that will allow us to make better use of the information in existing and upcoming observing systems, including SWOT.

From the experiments reported here, restricting the SWOT data to the inner half of the swath and applying observation averaging with a 5 km radius can reduce some of the problems caused by assimilating observations with large correlated errors, while retaining some of the benefits from the increased horizontal resolution. However, these choices will not be optimal for all systems and the precise level of averaging will likely depend on the resolution of the model used. To make full use in operational systems of the upcoming high spatial-resolution SSH observations from wide-swath altimeters, it will be necessary

to develop and test techniques to directly account for correlated errors within our data assimilation scheme.

## 7 Conclusions

The impact of assimilating simulated wide-swath altimetry observations from the upcoming SWOT mission was assessed using Observing System Simulation Experiments (OSSEs) with a high-resolution shelf-seas model. A Nature Run using different initial conditions and atmospheric forcing to the OSSEs provided the truth against which the OSSEs were assessed. Simulated observations with realistic errors were then sampled from the Nature Run and assimilated in the OSSEs. A Control run was performed assimilating the standard network of observations, including SST, T/S profiles and nadir altimeter SLA. Further experiments were performed assimilating the standard network of observations in addition to simulated SWOT observations: one with the full swath width and the full correlated errors; another with only half the width of the swath and the full correlated errors; and another experiment assimilating the full-width swath, but which included only uncorrelated errors in the simulated SWOT observations. Four further experiments were performed with the full and half swath data, but employing observation averaging with 5 km and 20 km averaging radii.

The aim of this work was to assess the impact of assimilating SWOT observations with and without the expected correlated errors and to assess simple methods to reduce problems associated with correlated errors. In the off-shelf regions, the SSH, surface currents, and subsurface temperature and salinity were improved to varying degrees by the assimilation of SWOT observations with uncorrelated errors, with both SSH and surface current RMS errors reduced by up to 20%. Similarly, the on-shelf temperature below the mixed layer and salinity at some depths were also improved. However, when correlated errors are included in the full swath SWOT observations, there is a degradation in the sub-surface temperature and salinity, and the SSH and surface currents are degraded with a clear increase in the mean surface currents. While restricting the SWOT data to the inner half of the swath and applying observation averaging with a 5 km radius negated most of the negative impacts in our system, it also severely limited the positive impacts.

This work was a first assessment of the potential of assimilating wide-swath altimeter observations in the Met Office shelf-seas analysis and forecasting system. By quantifying the impact of assimilating SWOT observations without correlated errors and the problems encountered when assimilating the observations with correlated errors in our current assimilation scheme, we hope to highlight the potential of swath altimeter observations and the challenge in their use within a current operational system.

The experiments presented here have shown that the assimilation of wide-swath altimetry observations has the potential to significantly improve the sea surface height and surface current errors in our operational analyses and forecasts. The assimilation of sub-surface profiles and altimetry can be complementary and so improve forecasts of the vertical structure of the ocean. This has the potential to bring benefits to our services to the marine industry and defence users, search and recovery operations, and others. With further investigation, the increased observation coverage offered by missions such as SWOT could also improve surge forecasting capabilities, potentially improving prediction during high impact weather events. To maximise the benefits from upcoming wide-swath altimetry missions, we aim to further explore methods to ameliorate the effects of correlated errors in the data assimilation and explore methods to explicitly model the correlated errors within our data assimilation scheme.

*Data availability.* The nature of the 4-D data generated in running the model experiments requires a large tape storage facility. These data are of the order of 10 TB (terabytes) for each of the ten simulations. The model data can be made available upon request from the authors.

*Author contributions.* RRK designed the experiments and their validation strategy. Both authors contributed to the scientific evaluation and analysis of the results.

*Competing interests.* The authors declare that they have no conflict of interest.

*Acknowledgements.* Funding support from the Copernicus Marine Environment Monitoring Service is gratefully acknowledged.

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
