# Peer review of "Assimilating realistically simulated wide-swath altimeter observations in a high-resolution shelf-seas forecasting system"

_Ocean Science, 2021_

## Author Response (AR1)

**RC1**

We thank the referee for their time and their helpful and constructive comments.

The referee's report is shown below (in bold) with our responses below each of their comments

The paper untitled « Assimilating realistically simulated wide-swath altimeter observations in a high-resolution shelf-seas forecasting system" present a very interesting and innovative study using state of the art methods and models and clearly presented. Main objectif of the study is to prepare assimilation of future wide-swath altimetry observation from SWOT satellite and to quantify impact and expected improvement of this future mission. Authors address this problem in a realistic operational high resolution ocean forecasting system and using an OSSE protocole perfectly defined and justified in term of ocean processes represented in the model and adequation with the model resolution, observations assimilated in the system complementarity of the observation data set and state of the art data assimilation method fully validated and used in an operational context.

The paper is well written, perfectly understandable, and well presented, plan of the paper is logical and help the reader. The information in each section is at the appropriate level.

The authors present important results obtain in this realistic context that are useful for scientists involved in the SWOT mission, for the developer of data assimilation method applied to oceanography and for operational ocean forecasting centers.

I fully recommend publication of the paper if authors can take into account the following remarks and recommendations.

General comments and questions

• the authors don't provide any figure of innovations or increments, they only show bias and RMSE. That should be justified in the text if analysis increments don't provide more information that the bias or RMSE. But I think that analysis increments provide important information to fully understand how the data assimilation scheme work. I expect that spatial scale of increments should be different and could be illustrated on figure 7 for example. Increments could be also useful to illustrate the discussion in section 5.2.2 and/or 5.2.3 on the improvement/degradation of the solution depending of assimilated observations.

Our analysis focussed on the bias and RMSE as with an OSSE we have the truth everywhere and so comparing the bias and RMSE between experiments gives a clear indication of the benefits/detriments of changes due to the observations assimilated. However, we agree that the analysis increments are useful in understanding how the assimilation scheme works and so we have updated Sections 5.1 & 5.3 to include a figure (new Figure 9) and discussion on the increments in the Control experiment compared to the LowErrSWOT experiment.

• One of the objectives of assimilating SWOT observations is to constrain smallscale structures in the ocean. This is not address in the paper (expect remark at the end of 5.3 without any illustration or explanation), the authors don't present any results illustrating the impact on meso scale structures in the different simulations or a spectral analysis presenting differences in term of energy between all the experiments. I understand that this is not the aim of the paper which is really focuses on the different sources of errors in the SWOT observations and especially the very important topic related to uncorrelated errors. I recommend adding at least a paragraph in the discussion section on the impact of SWOT on mesoscale structures and a perspective on this topic in the conclusion. Ideally, the authors will add a subsection in chapter 5 for example in section 5.1 about SSH.

As you say, the main aim of this paper was to investigate the limitations posed by correlated errors on the impact of SWOT. We intend to explore the use of a power spectral analysis in upcoming global OSSEs where the limited size of this regional model domain would not be an issue. However, the figure showing example daily increments, added in response the first comment, demonstrates that the assimilation scheme attempts to add more small-scale structures when swath altimetry is included. Additionally, in Section 5.3 we have attempted to demonstrate qualitatively the impact of assimilating SWOT observations on the surface currents. For example, Figure 15 shows improved errors in the surface currents at large- and small-scales suggesting the mesoscale structures are better initialised when assimilating the SWOT observations without correlated errors. Sections 5.1 & 5.3 and the discussion section have been updated to discuss this in more detail.

• The experimental protocol is well described and fully justified, especially with regard to how SWOT errors are represented in the system and the impact of these errors when the data are assimilated. The authors should provide recommendations in the discussion or conclusion section about how SWOT data should be post processed for an optimized use in data assimilation scheme. How could correlated errors be removed or reduced? Is the HalfSWOT or the 5km and 20km filtering solution a recommendation or a haddock solution? Is it realistic to expect that only kaRIn error will remain?

We have shown that using the inner half of the swath and median-averaging with a 5km radius can reduce many of the problems caused by the correlated errors in the simulated SWOT observations. However, these choices will not be optimal for all systems; the precise level of averaging will depend on the resolution of the model used. Although some preprocessing by the data providers may be possible, we are not the experts in this area and the current literature leads us to believe that correlated errors will be a significant issue when using near-real time swath altimetry from SWOT in operational systems. As mentioned in our discussion, we expect methods to account for correlated errors in the assimilation scheme will be necessary to fully exploit the observations from SWOT. Our discussion and conclusions have been updated to emphasise these aspects.  In Chapter 1 : Introduction. Authors could add a citation of recent publication Benkiran et al 2021 "Assessing the Impact of the Assimilation of SWOT Observations in a Global High-Resolution Analysis and Forecasting System Part 1: Methods"

References to this recent paper and the associated Part2 paper have been added. Thank you for the suggestion.

In section 3.1.1. the authors comment on an important point regarding the • differences between nature run and free run, in the OSSE protocol it is important to understand these differences and how the data assimilation scheme will move the model on another trajectory. In this section it is not clear why there is systematic cold and fresh bias. Is there a mistake in the explanation "due to broadly similar irradiative fluxes between the atmospheric forcing datasets". Is there a systematic bias between the two atmospheric forcings used in the experiment for the wind? the heat fluxes? The paper doesn't address the question of whether this systematic bias between nature run and free run have an impact on the results? Could you expect different impact on the sea level analysis in a unbiased system? The authors don't provide an OSSE calibration, comparing SLA differences between nature run and free run and what could be obtain in a real case assimilating real data. This is recommended to understand if in the OSSE experiment the data assimilation scheme will work as in a real case. I suggest to provide on fig 2 an additional map showing the classical SLA increment obtained in the operational system.

The different atmospheric forcing is one of the main methods we have used to introduce realistic differences between the Nature Run (NR) and OSSE runs. Both forcing sets are high-quality atmospheric forecasts. The difference between these forcing datasets is a fair reflection on the uncertainty in the true atmospheric forcing. The use of different surface forcing leads to changes in the ocean model mixed layer depth, which is the cause of the initial cold bias in the Free Run compared to the NR. This is readily corrected by SST assimilation (as shown in Figure 11). However, the initial fresh bias is primarily due to the different initial conditions used in the NR and OSSE runs. Again, this was deliberate to ensure the NR and OSSE runs had differences which reflect those between the real-world and our forecast systems. Both initial conditions used came from assimilative runs at the correct time of year and the differences reflect the uncertainty given the relative lack of sub-surface observations. Section 3.1.1 has been updated to clarify this.

To better understand if the SLA differences between the nature run and OSSE runs are similar to what might be expected in an operational system, we have compared the mean and RMS of the SSH increments from the Control and LowErrSWOT experiments with those from a separate experiment assimilating real observations in the same model and over the same time period. We found the bias were near zero in all cases and the RMS of the SSH increments was 1.21cm in the Control, 1.19cm in our experiment assimilating real observations, and 1.63cm in the LowErrSWOT experiment when simulated SWOT observations were included. We believe this demonstrates that our Control run is applying similar increments to an operational system assimilating real observations, and the simulate

SWOT observations allow more of the SSH variability to be observed and assimilated. Section 3.1.1 has been updated to discuss this.

• In section 4.2. It might be useful to provide a brief definition of each error and comment each figure 5 from a) to f). Could the authors provide more information on the following remark "The length-scale of these correlations can also be of the same order as the size of the domain". Is it something deduced from one of the figures? .

A brief description of each error has been added as suggested. Our comment on the length-scales was deduced from the figure – this has been clarified in the text.

• One important difference between Control run, SWOT and halfSWOT run is the number of sla observations in the system during each data assimilation cycle. The authors don't provide any information on the number of observations assimilated during an assimilation cycle and the expected impact when the data assimilation scheme assimilate half the observations.

We have updated Sections 4.1 & 4.2 with details on the average number of each observation type assimilated in each assimilation cycle. Each day, there are approximately 10^5 SST observations, 10 T/S profiles with ~1000 total observations, a few thousand nadir SLA observations, and 10^5 SWOT observations. The inclusion of SWOT therefore approximately doubles the total number of observations assimilated.

The number of iterations used (40) to minimise out 3D-var cost function was sufficient to reach a similar level of convergence in all our experiments, and so we do not expect the quantity of observations in itself to be a factor in the resulting impacts. Rather we have shown the impact of the increased spatial coverage from SWOT. The differences between the HalfSWOT and SWOT experiments (as discussed in Section 5) balances the effects of reducing the spatial coverage of swath altimeter observations while at the same time removing those with the largest correlated errors.

• In table 3, it is unclear how RMSE is computed. Is it computed in the observations space or in the model space? Only with the points where there are observations or for the full domain?

The RMSE values here have been calculated in model space over the full domain (and for the on- and off-shelf regions marked in Figure 1). This has been clarified in the table caption and in the main text in Section 5.

• In section 5.1, Authors noted considerable seasonal variation in the off-shelf SSH RMSE, but high frequency variability is even higher and not mentioned.

In Section 5.1, we had addressed the high-frequency RMSE variation seen in the on-shelf region, but had not commented specifically on the off-shelf RMSE. To a lesser extent, high frequency RMSE increases are also seen in the off-shelf RMSE time-series. This may be due to the misplacement of eddies: due to the distance and time between SWOT swaths, much of the mesoscale structures will still be unobserved. We have updated Section 5.1 to discuss this.

**• Fig 8 : Is the error computed with the same point for different experiments? Is it computed for all the points of the model grid?**

This figure shows the SSH RMSE calculated for the full off-shelf domain (as marked in Figure 1).

**• Section 5.1.1. the authors refer to fig 5 to explain that the correlation could be longer than 20km, which is not obvious on fig 5 as no correlation is shown.**

Figure 5 (now Fig. 6) shows the individual error contributions to SWOT observations on an example day. For the phase and roll errors in particular, the errors appear relatively constant along large portions of individual tracks which we have interpreted as long length-scale correlations.

**Comments on the form**

• Figure 2 : add "bottom panel" in the legend.

Done.

• Figure 3 : limit of color bar could be change for on-shelf temperature and salinity to highlight more detail on the figure

Done

• Table 2 : provide units

The table has been updated as the units were not displayed very prominently.

• Chapter 5. Section 5.1. Why don't the authors keep the same section structures with a separation between on and off-shelf for each variables? There is only one subsection in Section 5.1

For SSH and surface currents although the time-series separates the on- and off-shelf regions, the maps show the impacts over the full domain. We therefore felt that the on- and off-shelf regions could be best described in one section. For temperature and salinity, the additional

subsections were used to make the discussion easier to follow given the number of figures involved.

• Fig 8, 13 : It would be good to keep the same color code for all the figures and experiments. LowerrSWOT in purple for all the figure for example and used other color than blue, yellow for HalfSWOT and HalfSWOT5km.

The colours used for each experiment in Figures 8 and 13 (now Figs 10 & 16) differ to the remaining figures in part to allow an easy comparison of the subplots within those figures (i.e., comparing the effect of the 5km averaging on the HalfSWOT and SWOT experiments in the two panels). However, we agree that this could be improved, so we have changed the LowErrSWOT experiment to purple as in the other figures, and used different colours for the remaining experiments in Figures 8 & 13 to avoid confusion with other figures.

**• Fig 11. HalfSWOT\_5Km and SWOT experiments are reverse in the legend.**

Corrected.

**Two others comments/questions**

• The authors don't provide information on the observation errors used in the case of superobs (5km and/or 20km filtering). It could be useful to provide this information in table 2.

The observation errors used in the assimilation scheme were not changed in the experiments where SWOT observations were median-averaged. Although it may be beneficial to change the observation errors depending on the chosen level of averaging, the main aim of the averaging was to reduce the effect of the largest correlated errors rather than reducing the random component of the errors. We plan to make a more detailed examination of the impact of the observation errors in the future experiments. Section 5.1.1 has been updated to clarify this.

**• The authors don't provide information regarding significant wave hight used to compute the KaRin error. Is it the standard 2m swh that is used in this study?**

Yes, we used the default setting of 2m. We have updated Section 4.2 to highlight the limitations of SWOT observations in regions with larger SWHs.

**RC2**

We thank the referee for their time and their helpful and constructive comments.

The referee's report is shown below (in bold) with our responses below each of their comments.

 In the introductory preamble, present altimeter capabilities are a bit underestimated. On line 33: "Although along-track observations can have a sampling frequency of ~7 km, various sources of noise limit the feature resolution to 100 km (Xu and Fu, 2013)." This is pessimistic for modern altimeters. The small footprint of AltiKa, and the enhanced along-track resolution of the SAR-Mode Delayed Doppler altimeter on CryoSat, Sentinel-3A/B and Sentinel-6 has brought the resolved wavenumber spectrum down to ~50 km, or possibly less with advanced re-tracking (e.g., ALES – Passaro and Birol papers). Admittedly, this is along-track, and is not realized in 2-D gridded products.

Thank you for highlighting this. We have updated the text in Section 1 and changed the reference to the more recent analysis by Dufau et al. (2016).

2. Similarly, it is claimed (line 46) that "data from altimeters could help to constrain processes such as tides and storm surges which are represented in the model ... However, the sampling ... by existing nadir altimeters is not currently sufficient to adequately constrain them in the NWS region." I think it's more the case that we don't yet have DA schemes that can make full use of the observations in constraining these dynamical processes. Useful information is there in the data.

We agree that the data assimilation systems cannot yet make full use of these observations, but the sparse nature (in space and time) of current along-track altimeter observations compared to the surge in particular is problematic. We have updated the text in Section 1 to highlight the deficiency of existing DA schemes in this respect.

3. Are the open boundary conditions for Nature Run and OSSEs the same? It's not explicitly stated. One disturbing result that is never really explained is why there should be a slow drift in SSH bias. Is the free run model steadily changing net volume, that assimilation serves to restore by reimposing the MDT along with the observations?

Yes, the lateral boundaries for the Nature Run and the OSSEs come from the same sources – our 1/12 degree North Atlantic system and the CMEMS Baltic Sea forecasting system. We

have updated Section 2 to make this clear. Since the only differences between the Free and Nature Runs are the initial conditions and the surface forcing, we believe that there is a bias between the surface forcing datasets which is driving the slow drift in the SSH bias and that this is likely a difference in the evaporation-precipitation. It is known that the ECMWF IFS has a global average wet bias (https://doi.org/10.1175/JHM-D-20-0308.1) and the Met Office UM has been shown to have a wet bias in some regions (https://link.springer.com/article/10.1007/s12040-018-1023-3).

Although these biases cause a drift in the SSH, we believe that this adds a measure of realism into these idealised experiments. Our operational shelf-seas systems (a 7km and 1.5km system) use both of these sources of surface forcing and so such biases in the E-P will also be present in these systems.

4. I would be interested to see a map of the regions that are predominantly in the category of top-to-bottom temperature difference less than 2°C where the balance adjustments to temperature and salinity are not applied. This would add context to Figs. 7 and 11. Unfortunately, we are not offered a map view of the skill for temperature and salinity to complement Figs. 7 and 11, which is an oversight the authors might care to address in revision. I leave it to them to decide how to usefully present this 3-D skill assessment in a 2-D map.

A figure has been added to show the extent of this restriction on applying the SSH balancing changes to temperature and salinity for a summer and winter example day (new Figure 4). This figure is also referred to in later sections addressing your points (#7 & #8) below.

We have also added figures showing maps of the vertically averaged bias and RMSE for temperature and salinity which aid the interpretation of the profile statistics shown in later figures. This is addressed further in response to comments #7 & #8.

5. I have some reservations about how appropriate the balance operator approach is for shelf seas, but that's a can of worms we can't open here. However, I would not oppose some rampant speculation about how altimeter sea level data might be better exploited in shelf sea DA systems.

We fully agree that the existing balance relations within our data assimilation scheme are not optimised for the shelf-seas. We are currently developing global and shelf-seas ensemble systems with which we will be able to better represent both errors-of-the-day and the region-specific balances and length-scales. In the future, we plan to investigate the use of this information to adjust the balances and/or directly within a hybrid DA system combining ensemble information with climatological error covariances. We have expanded our discussion on the deficiencies of our existing balance relationship in Sections 5 & 6 to include the above information. 6. The term RMSE is not defined when it is first used, and it is not spelled out whether this is full Root Mean Squared Error of observation minus model, or what is frequently called Centered RMS Error in geophysics, being the RMS of the difference between observation *anomaly* and model *anomaly* from their respective means. CRMS and bias are independent errors. I suspect here we have CRMS, otherwise we would need to tease out the effect of bias in the RMSE statistics. But, conventionally, RMSE includes bias, so please clarify.

Here we have used Root Mean Squared Error. Throughout, the "error" part of the RMSE is the gridpoint-by-gridpoint difference between two model runs. Since we are running OSSEs, we know the "true" state everywhere (the Nature Run, NR) from which our simulated observations are drawn and so the error between each OSSE and the NR is not skewed by the observation sampling as would happen when comparing real-world observations (an incomplete and uncertain sample of the true state) to an operational system.

We have updated the text (on first use of RMSE in the abstract and at the start of Section 5) to clarify this.

7. I would welcome some speculation as to why temperature and salinity on the shelf is improved, but velocity and sea level are not. Here, some spatially explicit view of where the balance operator is being applied, and where it is not, might be instructive.

In stratified regions SLA observations contain information on vertical T/S structure which is effectively assimilated in the deeper ocean. In the stratified regions of the shelf, we can make similar positive adjustments. However, our assimilation scheme applies a ramping of the velocity balance near coasts which may be limiting the retention of these observations on-shelf and so their impact on the SSH and velocity statistics.

Additionally, our approach of using 25-hour mean fields in the simulation of the SLA observations and for the background in our innovations necessarily removes high-frequency signals which dominate on-shelf. When assimilation of the standard observations is introduced, the Control Run shows improved bias and RMSE relative to the Free Run with a very low RMSE of ~1cm on-shelf. There is little further improvement when assimilating additional observations. We intend to further explore adaptions to our assimilation scheme, including whether it would be beneficial to retain the higher-frequency signals in our innovations.

We have updated Section 6 to discuss these areas of ongoing investigation.

8. Indeed, I wonder if the results in Fig. 10 would differentiate further if they were conditionally averaged by whether the balance operator was applied, or not. I encourage this addition to the paper. Perhaps the authors already made this

**calculation and found it of no consequence. If so, a remark to that effect would be useful to readers.**

Thank you for the useful suggestion. We have investigated this further and found that in regions/periods where no balanced changes to temperature and salinity are applied when assimilating SSH, there is no appreciable difference between the various assimilating experiments. That is, while the Control Run is superior to the Free Run in terms of the temperature and salinity bias and RMSE, no further improvement is gained in these regions/times from assimilating additional SLA observations.

As mentioned earlier, we have added maps of the vertically-averaged bias and RMSE for temperature and salinity (new Figures 11 & 13) to demonstrate the spatial differences between the experiments and aid the interpretation of the spatially averaged profile statistics presented in Figures 12 & 14. In particular, these maps show that for the well-mixed regions in the Southern North Sea, the temperature profile is well-constrained by SST assimilation. We have also updated Section 5.2 and the on-shelf T/S profile figure to focus on the Northern North Sea during June-October during which time the full balanced changes are applied.

**9. Speed (Fig. 11) is only one measure of current errors. What about direction? The speed error could be zero but with respective currents pointing in opposite directions.**

While we agree that a more complete comparison would also include the current directions, we chose to compare the current speeds to allow a straight-forward interpretation of the effect of the assimilation changes. Maps showing the error in the current speed readily highlight where a particular experiment may be degraded or improved, while a similar map of the error in the current direction can be dominated by regions with very small currents.

**Minor comments:**

1. Fig. 2 and 3 captions. Please say whether the bias is Free minus Nature, or Nature minus Free.

These are both Free minus Nature. The figure captions have been updated to state this.

2. The resolution of many figures is poor. It looks to me like these are produced with matplotlib, in which the case the fix is simply to specify dpi resolution in savefig.

Thank you. Figure resolution has been increased.